# PERTURB AND LEARN: ENERGY-BASED MODELLING IN DISCRETE SPACES WITHOUT MCMC

## ABSTRACT

Energy-based models (EBMs) offer a flexible framework for probabilistic modelling across various data domains. However, training EBMs on discrete data poses significant challenges, primarily due to the intricacies of sampling in such spaces. In this work, we propose to train discrete EBMs with Energy Discrepancy which only requires the evaluation of the energy function at data points and their perturbed counterparts, thus eliminating the need for demanding sampling techniques like Markov chain Monte Carlo. Energy discrepancy offers theoretical guarantees applicable to a broad class of perturbation processes, of which we investigate three types: perturbations based on Bernoulli noise, deterministic transforms, and neighbourhood structures. We estimate the energy discrepancy loss effectively using importance sampling with two types of proposal distributions: uninformed and gradient-informed. Empirically, we demonstrate the efficacy of the proposed approaches in a wide range of applications, including Ising models training, discrete density estimation, graph generation, and discrete image modelling.

## 1 INTRODUCTION

Discrete structures are intrinsic to most types of data such as text, graphs, and images. Estimating the data generating distribution $p_{\text{data}}$ of discrete data sets with a probabilistic model can contribute greatly to downstream inference and generation tasks. Energy-based models (EBMs) are probabilistic models of the form $p_{\text{ebm}} \propto \exp(-U)$, where the flexible choice of the energy function $U$ allows great control in the modelling of different data structures. However, energy-based models are, by definition, unnormalised models and notoriously difficult to train due to the intractability of their normalisation, especially in discrete spaces.

Energy-based models are typically trained with the contrastive divergence (CD) algorithm (Hinton, 2002) which performs approximate maximum likelihood estimation by approximating the gradient of the log-likelihood with Markov Chain Monte Carlo (MCMC) techniques. This method has led to rich research results on sampling from discrete distributions to enable fast and accurate estimation of energy-based models (Zanella, 2020; Grathwohl et al., 2021; Zhang et al., 2022b; Sun et al., 2022b;a; 2023). However, the training of energy-based models with contrastive divergence remains challenging, as it relies on sufficiently fast mixing of Markov chains. Since accurate sampling from the EBM typically cannot be achieved, contrastive divergence lacks theoretical guarantees (Carreira-Perpinan & Hinton, 2005) and leads to biased estimates of the energy landscape (Nijkamp et al., 2019).

The recently introduced Energy Discrepancy (ED) (Schröder et al., 2023) is a new type of contrastive loss functional that, by definition, depends on neither gradients nor MCMC methods. Instead, the definition of ED only requires the evaluation of the energy function on positive and contrasting, negative samples which are generated by perturbing the data distribution. Currently, the work in Schröder et al. (2023) has two limitations: Firstly, it only relies on Gaussian perturbations, limiting the approach to continuous settings. Secondly, the variance of the contrastive potential can not be controlled, forcing the practitioner to use a relatively small noise scale which provably limits the expressiveness of energy discrepancy.

In this work, we propose a framework to train energy-based models with energy discrepancy on discrete data. Our approach offers both flexibility and theoretical guarantees in the construction of negative samples, of which we mainly consider three types of perturbation based on Bernoulli

noise, deterministic transform, and neighbourhood structures. We then relate energy discrepancy to importance sampling, which not only provides a novel interpretation of contrastive samples used in the construction of the loss function, but also a new approach to constructing negative samples using gradient information. Finally, we demonstrate the effectiveness of energy discrepancy training on various discrete estimation tasks and show that energy discrepancy scales gracefully to high-dimensional image datasets and graphs.

## 2 ENERGY DISCREPANCIES

Energy-based models (EBMs) are a parametric family of distributions $p_\theta$ defined as

$$p_\theta(\mathbf{x}) = \frac{\exp(-U_\theta(\mathbf{x}))}{Z_\theta}, \quad Z_\theta = \sum_{\mathbf{x} \in \mathcal{X}} \exp(-U_\theta(\mathbf{x})), \tag{1}$$

where $U_\theta$ is the energy function parameterised by $\theta$ and $Z_\theta$ denotes the normalisation constant. In this work, we restrict $\mathcal{X}$ to a discrete space and mainly consider it to be $\{0,1\}^d$. Given a set of *i.i.d.* samples $\{\mathbf{x}^i\}_{i=1}^N$ from an unknown data distribution $p_{\text{data}}(\mathbf{x})$, our goal is to learn an EBM $p_\theta(\mathbf{x})$ to approximate $p_{\text{data}}(\mathbf{x})$. The *de facto* standard for finding such $\theta$ is to minimise the negative log-likelihood of $p_\theta$ under the data distribution via gradient decent

$$-\nabla_\theta \mathbb{E}_{p_{\text{data}}(\mathbf{x})}[\log p_\theta(\mathbf{x})] = \mathbb{E}_{\mathbf{x} \sim p_{\text{data}}}[\nabla_\theta U_\theta(\mathbf{x})] - \mathbb{E}_{\mathbf{x}_- \sim p_\theta}[\nabla_\theta U_\theta(\mathbf{x}_-)]. \tag{2}$$

The intuition behind this update is to decrease the energy of positive data samples $\mathbf{x} \sim p_{\text{data}}(\mathbf{x})$ and to increase the energy of negative samples $\mathbf{x}_- \sim p_\theta(\mathbf{x})$. However, the computation of gradient in (2) is known to be NP-hard in general (Jerrum & Sinclair, 1993). Consequently, existing approaches resort to sampling from the model $p_\theta$ to approximate the gradient of log-likelihood via Monte Carlo estimation. In discrete settings, the most popular sampling methods include the locally informed sampler (Zanella, 2020), Gibbs with gradients (GwG) (Grathwohl et al., 2021), discrete Langevin (Zhang et al., 2022b), and generative flow networks (GFlowNet) (Zhang et al., 2022a). Despite their established success in discrete energy-based modelling, these methods necessitate a trade-off that hampers scalability: running the sampler for an extended duration escalates the cost of maximum likelihood training, while shorter sampler runs yield inaccurate approximations of the likelihood gradient and introduce biases into the learned energy.

Energy Discrepancy (Schröder et al., 2023) is a recently proposed method to train energy-based models without the need for an extensive sampling process. Instead, it constructs negative samples by perturbing the data, thus bypassing the sampling step while still yielding a valid training objective. To elucidate, the energy discrepancy is formally defined as follows:

**Definition 1** (Energy Discrepancy). *Let $p_{\text{data}}$ be a positive density on a measure space $(\mathcal{X}, \mathrm{d}\mathbf{x})$[1] and let $q(\mathbf{y}|\mathbf{x})$ be a conditional probability density. Define the* contrastive potential *induced by $q$ as*[2]

$$U_q(\mathbf{y}) := -\log \sum_{\mathbf{x}' \in \mathcal{X}} q(\mathbf{y}|\mathbf{x}') \exp(-U(\mathbf{x}')) \tag{3}$$

*We define the* energy discrepancy *between $p_{\text{data}}$ and $U$ induced by $q$ as*

$$\mathrm{ED}_q(p_{\text{data}}, U) := \mathbb{E}_{p_{\text{data}}(\mathbf{x})}[U(\mathbf{x})] - \mathbb{E}_{p_{\text{data}}(\mathbf{x})}\mathbb{E}_{q(\mathbf{y}|\mathbf{x})}[U_q(\mathbf{y})]. \tag{4}$$

The validity of this loss functional is given by the following non-parametric estimation result:

**Proposition 1** (Schröder et al. (2023)). *Let $p_{\text{data}}$ be a positive probability density on $(\mathcal{X}, \mathrm{d}\mathbf{x})$. Under mild conditions on $q$, energy discrepancy $\mathrm{ED}_q$ is functionally convex in $U$ and has, up to additive constants, a unique global minimiser $U^* = \arg\min \mathrm{ED}_q(p_{\text{data}}, U)$. Furthermore, this minimiser is the Gibbs potential for the data distribution, i.e. $p_{\text{data}} \propto \exp(-U^*)$.*

Beyond non-parametric estimation, the validity of energy discrepancy can be understood from other perspectives. Specifically, the loss function defined in (4) is equivalent to the expected negative log-likelihood of the posterior $p_{\text{ebm}}(\mathbf{x}|\mathbf{y})$

$$\arg\min_U \mathrm{ED}_q(p_{\text{data}}, U) \Leftrightarrow \arg\min_U -\mathbb{E}_{p_{\text{data}}(\mathbf{x})}\mathbb{E}_{q(\mathbf{y}|\mathbf{x})}[\log p_{\text{ebm}}(\mathbf{x}|\mathbf{y})] \tag{5}$$

---

[1]On discrete spaces $\mathrm{d}\mathbf{x}$ is assumed to be a counting measure. On continuous spaces $\mathcal{X}$, the appearing sums and expectations turn into integrals with respect to the Lebesgue measure

[2]With a slight abuse of notations, we represent the contrastive potential induced by distribution $q$ as $U_q$ and denote the energy function as $U$ with or without the subscript $\theta$.

where $p_{\text{ebm}}(\mathbf{x}|\mathbf{y}) \propto \exp(-U(\mathbf{x}))q(\mathbf{y}|\mathbf{x})$. Furthermore, minimising energy discrepancy is also equivalent to minimising the KL-contraction divergence (Lyu, 2011; Luo et al., 2023)

$$\underset{U}{\arg\min}\, \text{ED}_q(p_{\text{data}}, U) \Leftrightarrow \underset{U}{\arg\min}\, \text{KLC}_{\mathcal{Q}}(p_{\text{data}}, p_{\text{ebm}}), \quad p_{\text{ebm}} \propto \exp(-U), \qquad (6)$$

where $\text{KLC}_{\mathcal{Q}}(p_1 \| p_2) := \text{KL}(p_1 \| p_2) - \text{KL}(\mathcal{Q}p_1 \| \mathcal{Q}p_2)$ denotes the KL-contraction divergence and $\mathcal{Q}p(\mathbf{y}) := \sum_{\mathbf{x} \in \mathcal{X}} q(\mathbf{y}|\mathbf{x})p(\mathbf{x})$ is a convolution operator. Notably, $\text{KLC}_{\mathcal{Q}}(p_1, p_2)$ is non-negative and equals zero if and only if $p_1 = p_2$, *a.e.*. Energy discrepancy has demonstrated notable effectiveness in training EBMs in continuous spaces (Schröder et al., 2023). In the next section, we take this concept a step further by applying it to the training of EBMs in discrete spaces.

## 3 ENERGY DISCREPANCIES FOR DISCRETE DATA

To apply energy discrepancy in the training of energy-based models in discrete spaces, it is critical to select a suitable discrete perturbation $q(\mathbf{y}|\mathbf{x})$, and establish an effective approach for estimating the contrastive potential $U_q$. In this section, we begin by introducing various types of perturbations and then delve into the estimation of $U_q$ using importance sampling. We will focus on binary discrete data, i.e. $\mathcal{X} = \{0, 1\}^d$, and discuss possibilities for extensions to other types of discrete data briefly.

### 3.1 VARIANTS OF DISCRETE PERTURBATION

As per Proposition 1, the perturbation $q(\mathbf{y}|\mathbf{x})$ can be chosen quite generally as long as it can be guaranteed that computing $\mathbf{y}$ comes at a loss of information. In the following, we introduce three categories for constructing such perturbative processes.

**Bernoulli Perturbation.** As proposed previously in (Schröder et al., 2023, Appendix B.3), $q$ can be defined via a Bernoulli distribution, i.e. the Bernoulli perturbed data point is obtained as $\mathbf{y} = \mathbf{x} + \boldsymbol{\xi}$ mod 2 for $\boldsymbol{\xi} \sim \text{Bernoulli}(\varepsilon)^d, \varepsilon \in (0, 1)$. This induces a symmetric transition density $q(\mathbf{y}|\mathbf{x})$ on $\{0, 1\}^d$. The Bernoulli random variable $\boldsymbol{\xi}_k$ indicates in each dimension whether to flip the entry of $\mathbf{x}$. The value of $\epsilon$ controls the information loss induced by the perturbation. In theory, larger values of $\epsilon$ lead to a more data-efficient loss, while smaller values of $\epsilon$ may be more practical as they contribute to improved training stability. On discrete spaces with more than two states per dimension, this perturbation can be generalised to a Markov transition density on the state space $\{0, \ldots, K - 1\}^d$.

**Deterministic Transformation.** The perturbation $q$ can also be defined through a deterministic information loosing map. Specifically, consider a mapping $g : \mathcal{X} \to \mathcal{Y}$, which is not injective at any $\mathbf{x} \in \mathcal{X}$, *i.e.,* for all $\mathbf{y} \in \mathcal{Y}$, we have $|\{\mathbf{x} \in \mathcal{X} : g(\mathbf{x}) = \mathbf{y}\}| > 1$. Then, the characteristic function $q(\mathbf{y}|\mathbf{x}) = \delta_{\{g(\mathbf{x})\}}(\mathbf{y})$ is a suitable perturbation, where $\delta$ denotes the indicator function[3]. It is noteworthy that such transformations offer flexibility, and common augmentation techniques (Zhao et al., 2020), like pooling, resizing, cutout, etc., are suitable options. In this paper, we primarily focus on mean pooling (details in Appendix B.1), but exploring more tailored transformations for specific forms of data is a potential avenue for future research.

**Neighbourhood-based Perturbation.** Inspired by concrete score matching (Meng et al., 2022), we introduce the last perturbation scheme based on neighbourhood maps: $\mathbf{x} \mapsto \mathcal{N}(\mathbf{x})$, which assigns each data point $\mathbf{x} \in \mathcal{X}$ a set of neighbours $\mathcal{N}(\mathbf{x})$. In this case, the forward transition density is given by the uniform distribution over the set of neighbours, *i.e.,* $q(\mathbf{y}|\mathbf{x}) = \frac{1}{|\mathcal{N}(\mathbf{x})|}\delta_{\mathcal{N}(\mathbf{x})}(\mathbf{y})$. In this work, we mainly consider the grid neighbourhood, which is constructed as

$$\mathcal{N}_{\text{grid}}(\mathbf{x}) = \{\mathbf{y} \in \{0, 1\}^d : \mathbf{y} - \mathbf{x} = \pm \mathbf{e}_k, k = 1, 2, \ldots, d\}, \qquad (7)$$

where $\mathbf{e}_k$ is a vector of zeros with a one in the $k$-th entry. Notably, this neighbourhood structure also exhibits symmetry, *i.e.,* $\mathcal{N}_{\text{grid}}^{-1}(\mathbf{x}) = \mathcal{N}_{\text{grid}}(\mathbf{x})$, which will make the computation of contrastive potential particularly simple. Neighbourhood structures are particularly suitable for adaptation to other types of discrete data as they offer great modelling flexibility.

### 3.2 ESTIMATING CONTRASTIVE POTENTIAL WITH IMPORTANCE SAMPLING

The primary challenge in turning energy discrepancy into a practical loss function lies in the estimation of the contrastive potential $U_q$. To make the estimation tractable, we first turn the intractable sum

---

[3]The indicator function $\delta_{\mathcal{S}}(\mathbf{x})$ equals 1 if $\mathbf{x} \in \mathcal{S}$ else 0 if $\mathbf{x} \notin \mathcal{S}$.

over the whole state space $\mathcal{X}$ into an expectation using importance sampling, *i.e.,*

$$U_q(\mathbf{y}) = -\log \sum_{\mathbf{x} \in \mathcal{X}} \exp(-U(\mathbf{x}))q(\mathbf{y}|\mathbf{x}) = -\log \mathbb{E}_{\rho_{\mathbf{y}}(\mathbf{x})}\left[w_{\mathbf{y}}(\mathbf{x})\exp(-U(\mathbf{x}))\right], \qquad (8)$$

where $\rho_{\mathbf{y}}(\mathbf{x})$ is a conditional distribution in $\mathbf{x}$, and $w_{\mathbf{y}}(\mathbf{x}) := \frac{q(\mathbf{y}|\mathbf{x})}{\rho_{\mathbf{y}}(\mathbf{x})}$ denotes the importance weight. The minimum-variance proposal takes the form of the posterior distribution (Robert et al., 1999)

$$\rho_{\mathbf{y}}^*(\mathbf{x}) \propto \exp(-U(\mathbf{x}))q(\mathbf{y}|\mathbf{x}),. \qquad (9)$$

We need to make a trade-off between finding tractable approximations of $\rho_{\mathbf{y}}^*$ and the computational complexity of our approach. Ideally, one chooses large perturbations such as the Bernoulli perturbation that are capable of perturbing multiple dimensions at once and explore the state space effectively, yielding a data efficient loss function. Since this makes the estimated loss more noisy, an ideal proposal distribution $\rho_{\mathbf{y}}$ should be easy to sample from and yield a low variance estimator.

### 3.3 Uninformed Proposals

If the perturbing distribution $q$ can be normalised in $\mathbf{x}$, a naive approach is to replace $\rho_{\mathbf{y}}^*$ with

$$\pi_{\mathbf{y}}(\mathbf{x}) := \frac{q(\mathbf{y}|\mathbf{x})}{\sum_{\mathbf{x} \in \mathcal{X}} q(\mathbf{y}|\mathbf{x})}.$$

We call this the **uninformed proposals**, which involve replacing the energy-based factor of $\rho^*$ with a uniform distribution, implying that no information about $U$ is utilised in estimating $U_q$. In this case, the importance weight only contributes a constant to the contrastive potential independent of $U$. Schröder et al. (2023) utilise this approach to approximate the energy discrepancy loss based on Gaussian perturbations which are symmetric in $\mathbf{y}$ and $\mathbf{x}$ and are thus normalised by definition.

In discrete settings, the corresponding uninformed proposals $\pi_{\mathbf{y}}$ for the three types of proposed perturbation have simple forms and allow easy approximations of the contrastive potential. For the Bernoulli perturbation, the likelihood $q$ is symmetric in $\mathbf{x}$ and $\mathbf{y}$ and thus already normalised, *i.e.,* $\pi_{\mathbf{y}}(\mathbf{x}) = \prod_{k=1}^d \epsilon^{\boldsymbol{\xi}_k}(1-\epsilon)^{1-\boldsymbol{\xi}_k}$ with $\boldsymbol{\xi} := |\mathbf{y} - \mathbf{x}| \in \{0,1\}^d$. Hence, we can approximate expectations by sampling Bernoulli noise and flipping entries of the data vector accordingly, i.e.

$$\mathbf{x}_- := (\mathbf{y} + \boldsymbol{\xi} \mod 2) \sim \pi_{\mathbf{y}}(\mathbf{x}) \quad \text{where} \quad \boldsymbol{\xi} \sim \text{Bernoulli}(\epsilon). \qquad (10)$$

For the deterministic transformation $\mathbf{y} = g(\mathbf{x})$, the appropriate normalised proposal distribution $\pi_{\mathbf{y}}$ is obtained by sampling wuniformly from the preimage of $g$, i.e. $\mathbf{x}_- \sim \text{Uniform}(\{\mathbf{x} \in \mathcal{X} : g(\mathbf{x}) = \mathbf{y}\})$. In this case, the importance weight is given by $w_{\mathbf{y}} := |\{g^{-1}(\mathbf{y})\}|$ which is independent of $U$. The grid-neighbourhood case can be treated similarly. Here, the contrastive potential can be expressed in terms of the inverse neighbourhood $\mathbf{y} \mapsto \mathcal{N}^{-1}(\mathbf{y})$, *i.e.,* by sampling uniformly from the set of points that have $\mathbf{y}$ to their neighbour. In all the above cases, the contrastive potential can finally be approximated by sampling from $\pi_{\mathbf{y}}(\mathbf{x})$ which yields

$$U_q(\mathbf{y}) \approx \log \frac{1}{M} \sum_{i=1}^M \exp(-U(\mathbf{x}_-^i)) + \log w_{\mathbf{y}}, \quad \mathbf{x}_-^i \sim \pi_{\mathbf{y}}(\mathbf{x}). \qquad (11)$$

## 4 Improvement with Gradient Guidance and Stabilisation

While utilising the uninformed proposal $\pi_{\mathbf{y}}$ can effectively estimate the contrastive potential, it is desirable to reduce its variance through a better approximation of the optimal potential $\rho_{\mathbf{y}}^*$. In this section, we tackle this issue by initially introducing gradient-informed proposals, drawing inspiration from Grathwohl et al. (2021), and subsequently enhancing training stability through the use of $w$-stabilisation as proposed by Schröder et al. (2023).

### 4.1 Gradient-informed Proposals

For many choices of perturbation, such as Bernoulli and Grid-neighbourhood perturbations, the distribution $q(\mathbf{y}|\mathbf{x})$ only depends on the difference between $\mathbf{x}$ and $\mathbf{y}$. Sampling from the proposal

$\rho_{\mathbf{y}}(\mathbf{x})$ can be simplified when $\rho_{\mathbf{y}}(\mathbf{x})$ also only depends on this difference. Inspired by the usage of gradients in Grathwohl et al. (2021); Liu et al. (2023), we suggest achieving this via a Taylor expansion $U(\mathbf{x}) \approx U(\mathbf{y}) + \nabla U(\mathbf{y})^T(\mathbf{x} - \mathbf{y})$. This leads to the **gradient-informed proposals**, which is a first-order approximation to the optimal proposal $\rho_{\mathbf{y}}^*$ in (9)

$$\rho_{\mathbf{y}}(\mathbf{x}) \propto \exp\left(-\frac{1}{\tau}\nabla U(\mathbf{y})^T(\mathbf{x} - \mathbf{y})\right) q(\mathbf{y}|\mathbf{x}). \tag{12}$$

Here, $U(\mathbf{y})$ was absorbed into the normalisation of the proposal, and we introduce a temperature parameter $\tau$ to control the sharpness of the proposal. Obviously, when $\tau \to \infty$, the gradient-informed proposal converges to the uninformed proposal. When $\tau \to 0$, the proposal only tries to minimise the linearised energy landscape which may miss the information of local low-energy states. Zanella (2020) suggests using $\tau = 2$ to balance the two effects for locally balanced proposals, which was also used in Grathwohl et al. (2021) for the gradient-informed proposal.

Note that the Taylor series is technically not well-defined for discrete data $\mathbf{x} \in \{0, 1\}$. However, the inner product $\nabla U(\mathbf{y})^T(\mathbf{x} - \mathbf{y})$ is still a meaningful approximation of the difference $U(\mathbf{x}) - U(\mathbf{y})$ and can expressed as $\nabla U(\mathbf{y})^T(\mathbf{x} - \mathbf{y}) = (\nabla U(\mathbf{y}) \odot (1 - 2\mathbf{y}))^T \boldsymbol{\xi}$, where $\odot$ denotes a dimension-wise multiplication and $\boldsymbol{\xi} = |\mathbf{x} - \mathbf{y}| \in \{0, 1\}^d$. To abbreviate the notation, we write $\nabla^{\text{bit}} U(\mathbf{y}) := (\nabla U(\mathbf{y}) \odot (1 - 2\mathbf{y}))$ for the gradient in $\{0, 1\}^d$. In the Bernoulli case, the gradient-informed proposal is again Bernoulli with parameter

$$p_k := \frac{\exp(-\frac{1}{\tau}\nabla_k^{\text{bit}} U(\mathbf{y}))\epsilon}{1 - \epsilon + \exp(-\frac{1}{\tau}\nabla_k^{\text{bit}} U(\mathbf{y}))\epsilon} \quad \text{for} \quad k = 1, 2, \ldots, d \tag{13}$$

*i.e.*, the proposal takes the form $\rho_{\mathbf{y}}^{\nabla}(\mathbf{x}) = \prod_{k=1}^d p_k^{\xi_k}(1 - p_k)^{1 - \xi_k}$. In the case of grid-neighbourhood perturbation, we flip at exactly one dimension which is determined via the categorical distribution

$$\mathbf{y}_{\neg k} \sim \rho_{\mathbf{y}}^{\nabla}(\mathbf{x}) \quad \text{where} \quad k \sim \text{Categorical}\left(\text{softmax}\left(-\frac{1}{\tau}\nabla^{\text{bit}} U(\mathbf{y})\right)\right), \tag{14}$$

where $\mathbf{y}_{\neg k} = \mathbf{y} + \mathbf{e}_k \mod 2$ denotes the vector $\mathbf{y}$ with the entry in the $k$th dimension flipped. In both cases, the calculation of the importance weight reduces to $w_{\mathbf{y}}(\mathbf{x}) \propto \exp\left(\frac{1}{\tau}\nabla^{\text{bit}} U(\mathbf{y})^T \boldsymbol{\xi}\right)$. The proportionality constant is given by the normalisation of $\rho_{\mathbf{y}}^{\nabla}(\mathbf{x})$ which is independent of the negative samples and hence does not influence the direction of the parameter gradient. We give derivations for the gradient-informed proposals in Appendix A.

**Comparison to CD-1.** In the special case of one negative sample per data point, the importance weight does not influence the gradient of energy discrepancy and the parameter update induced by one data point takes the form $\nabla_\theta U_\theta(\mathbf{x}) - \nabla_\theta U_\theta(\mathbf{x}_-)$ with $\mathbf{x}_- \sim \rho_{\mathbf{y}}^{\nabla}(\mathbf{x})$. For the grid neighbourhood, this update closely resembles the parameter update of contrastive divergence with one step of Gibbs with gradients (Grathwohl et al., 2021), with the primary difference that the sampler is initialised at $\mathbf{y} \sim q(\mathbf{y}|\mathbf{x})$ instead of $\mathbf{x}$. However, energy discrepancy provides a framework to increase the number of negative samples per data point and incorporate importance weights $w_{\mathbf{y}}(\mathbf{x}_-)$. In the limit, we approximate $\text{ED}_q(p_{\text{data}}, E_\theta)$ (see Theorem 1), which offers theoretical guarantees. Consequently, energy discrepancy with gradient-informed proposals has the capacity to learn more accurate energy landscapes than CD-1. This assertion is further substantiated by empirical evidence presented in Table 6, where we observe that energy discrepancy significantly outperforms CD-1.

## 4.2 TRAINING DISCRETE EBMS WITH ENERGY DISCREPANCIES

The above schemes permit the approximation of the contrastive potential from $M$ samples which are generated by first sampling $\mathbf{y} \sim q(\mathbf{y}|\mathbf{x})$, after which we compute $M$ approximate recoveries $\{\mathbf{x}_-^j\}_{j=1}^M$. The full loss can then be constructed for each data point $\mathbf{x} \sim p_{\text{data}}$ by calculating $\log \sum_{j=1}^M \exp(U(\mathbf{x}) - U(\mathbf{x}_-^j) + \log w_{\mathbf{y}}(\mathbf{x}_-^j)) - \log(M)$ using the numerically stabilised logsumexp function. However, this estimator for energy discrepancy is biased due to the logarithm and can exhibit high variance. As in Schröder et al. (2023), we stabilise training by introducing an offset for the logarithm which introduces a deterministic lower bound for the loss. This yields the energy discrepancy loss function

---

**Algorithm 1** Training Discrete EBMs with Energy Discrepancies

---

**Input**: Training data $\mathcal{D} = \{\mathbf{x}^i\}_{i=1}^N$; parameterised energy function $U_\theta$; perturbation dist. $q$; number of negative samples $M$; stabilisation parameter $w$.

1: **repeat**
2:     Sample a batch of data $\{\mathbf{x}^i\} \sim \mathcal{D}$
3:     Sample perturbed data $\mathbf{y}^i \sim q(\mathbf{y}|\mathbf{x}^i)$ for each $\mathbf{x}^i$
4:     Compute the gradient $\nabla_{\mathbf{y}^i} U_\theta(\mathbf{y}^i)$ if using gradient-informed proposals
5:     Sample $M$ negative samples $\{\mathbf{x}_-^{i,j}\}_{j=1}^M$ for each $\mathbf{x}^i$ from proposals in (9), (13) or (14)
6:     Update $\theta$ based on $\nabla_\theta \mathcal{L}_{q,M,w}(U_\theta)$ via (15)
7: **until** convergence of parameter $\theta$

---

$$\mathcal{L}_{q,M,w}(U) := \frac{1}{N} \sum_{i=1}^N \log\left(w + \sum_{j=1}^M \exp(U(\mathbf{x}^i) - U(\mathbf{x}_-^{i,j}) + \log w_{\mathbf{y}^i}(\mathbf{x}_-^{i,j}))\right) - \log(M) \quad (15)$$

with $\mathbf{x}^i \sim p_{\text{data}}$, $\mathbf{y}^i \sim q(\cdot|\mathbf{x}^i)$, and $\mathbf{x}_-^{i,j} \sim \pi_{\mathbf{y}^i}$ or $\rho_{\mathbf{y}^i}$. It is worth noting that the importance weight $w_{\mathbf{y}^i}(\mathbf{x}_-^{i,j})$ remains constant when employing the uninformed proposal $\pi_{\mathbf{y}^i}$ and can therefore be omitted during optimisation. Alternatively, if using the gradient-informed proposals, $w_{\mathbf{y}^i}(\mathbf{x}_-^{i,j})$ can be explicitly computed due to their closed-form expressions defined in (13) and (14). Notably, as shown in the following theorem, this approximation is consistent for any fixed $w$:

**Theorem 1.** *For every $\varepsilon > 0$ there exist $N, M \in \mathbb{N}$ such that $|\mathcal{L}_{q,M,w}(U) - \text{ED}_q(p_{\text{data}}, U)| < \varepsilon$ a.s..*

We adapted the proof of Theorem 1 to the case of discrete perturbations and gradient proposals in Appendix B.2. The training procedure is outlined in Algorithm 1, where we present five distinct approaches: i) ED-Bern with Bernoulli perturbation and uninformed proposal; ii) ED-Pool with mean-pooling transformation and uninformed proposal; iii) ED-Grid with grid-neighbourhood perturbation and uninformed proposal; iv) ED-∇Bern with Bernoulli perturbation and gradient-informed proposal; and v) ED-∇Grid with grid-neighbourhood perturbation and gradient-informed proposal.

## 5   RELATED WORK

**Contrastive loss functions.** Our work is based on energy discrepancies first introduced in (Schröder et al., 2023). Energy discrepancy is equivalent to certain types of KL contraction divergences which were already introduced in Lyu (2011), however, only for its theoretical properties. Interestingly, the structure of the stabilised energy discrepancy loss shares similarities with other contrastive losses such as Ceylan & Gutmann (2018); Gutmann & Hyvärinen (2010); van den Oord et al. (2018). This poses the question of possible classification-based interpretations of energy discrepancy and of the $w$-stabilisation.

**Contrastive divergence and Sampling.** Contrastive divergence is commonly utilized for training energy-based models in continuous spaces with Langevin dynamics (Xie et al., 2016; 2018; 2022; Du et al., 2020; Xiao et al., 2020). In discrete spaces, EBM training heavily relies on contrastive divergence methods as well, driving extensive exploration and development in the realm of discrete sampling strategies. The improvement of the standard Gibbs method was proposed by Zanella (2020) through locally informed proposals. This method was extended to include gradient information (Grathwohl et al., 2021) to drastically reduce the computational complexity of flipping bits of binary valued data and to flipping bits in several places (Sun et al., 2022b; Emami et al., 2023; Sun et al., 2022a). Moreover, the discrete version of Langevin sampling have been introduced based on this idea (Zhang et al., 2022b; Rhodes & Gutmann, 2022; Sun et al., 2023). Consequently, most current implementations of contrastive divergence use multiple steps of a gradient-based discrete sampler. Alternatively, energy-based models can be trained using generative flow networks which learn a Markov chain to construct data by optimising a given reward function. The Markov chain can be used to obtain samples for contrastive divergence without MCMC from the EBM (Zhang et al., 2022a).

**Other training methods for discrete EBMs.** There also exist some MCMC-free approaches for training discrete EBMs. Our work has connections to concrete score matching (Meng et al., 2022)

Table 1: Experimental results of discrete density estimation. We display the negative log-likelihood (NLL). The results of baselines are taken from Zhang et al. (2022a).

| Metric | Method | 2spirals | 8gaussians | circles | moons | pinwheel | swissroll | checkerboard |
|--------|--------|----------|-----------|---------|--------|----------|-----------|--------------|
| NLL↓ | PCD | 20.094 | 19.991 | 20.565 | 19.763 | 19.593 | 20.172 | 21.214 |
| | ALOE+ | 20.062 | 19.984 | 20.570 | 19.743 | 19.576 | 20.170 | 21.142 |
| | EB-GFN | 20.050 | 19.982 | **20.546** | 19.732 | 19.554 | 20.146 | 20.696 |
| | ED-Bern | **20.039** | 19.992 | 20.601 | **19.710** | 19.568 | **20.084** | 20.679 |
| | ED-∇Bern | 20.048 | 19.979 | 20.603 | 19.717 | **19.553** | 20.089 | **20.677** |
| | ED-Grid | 20.049 | **19.965** | 20.601 | 19.715 | 19.564 | 20.088 | 20.678 |
| | ED-∇Grid | 20.092 | 20.005 | 20.605 | 19.740 | 19.577 | 20.087 | 21.439 |

through the usage of neighbourhood structures to define a replacement of the continuous score function. Another sampling-free approach for training discrete EBMs is ratio matching (Hyvärinen, 2007; Lyu, 2012). It has been found that gradient information drastically improves the performance of ratio matching as well (Liu et al., 2023). Moreover, Dai et al. (2020) proposed to apply variational approaches to train discrete EBMs instead of MCMC. Eikema et al. (2022) replaced the widely-used Gibbs algorithms with quasi-rejection sampling to trade off the efficiency and accuracy of the sampling procedure. The perturb-and-map (Papandreou & Yuille, 2011) is also recently utilised to sample and learn in discrete EBMs (Lazaro-Gredilla et al., 2021).

## 6 EXPERIMENTS

We demonstrate the efficacy of our methods on various tasks, including training Ising models, density estimation on a discretised 2D plane, graph generation, and discrete image modelling. Here we mainly showcase the empirical results of ED-Bern, ED-∇Bern, ED-Grid, and ED-∇Grid, but leave the results of ED-Pool on density estimation and more implementation details in Appendix C.

### 6.1 TRAINING ISING MODELS

We first evaluate our methods on training the lattice Ising model, which has the form of

$$p(\mathbf{x}) \propto \exp(\mathbf{x}^T J \mathbf{x}), \ \mathbf{x} \in \{-1, 1\}^D,$$

where $J = \sigma A_D$ with $\sigma \in \mathbb{R}$ and $A_D$



Figure 1: Results on learning Ising models. Left to right: ground truth, ED-Bern, ED-∇Bern, ED-Grid, ED-∇Grid.

being the adjacency matrix of a $D \times D$ grid. Following Grathwohl et al. (2021); Zhang et al. (2022b;a), we generate training data through Gibbs sampling and use the generated data to fit a symmetric matrix $J$ via energy discrepancy. Note that the training algorithms do not have access to the data-generating matrix $J$, only to the collection of samples. In Figure 1, we consider $D = 10 \times 10$ grids with $\sigma = 0.2$ and illustrate the learned matrix $J$ using a heatmap. It can be seen that the variants of energy discrepancy can identify the pattern of the ground truth, confirming the effectiveness of our methods. We defer experimental details and quantitative results comparing with baselines to Appendix C.1.

### 6.2 DISCRETE DENSITY ESTIMATION

In this experiment, we follow the experimental setting of Dai et al. (2020); Zhang et al. (2022a), which aims to model discrete densities over 32-dimensional binary data that are discretisations of continuous densities on the plane (see the top row in Figure 2). Specifically, we convert each planar data point $\hat{\mathbf{x}} \in \mathbb{R}^2$ to a binary data point $\mathbf{x} \in \{0, 1\}^{32}$ via Gray code (Gray, 1953). Consequently, the models face the challenge of modelling data in a discrete space, which is particularly difficult due to the non-linear transformation from $\hat{\mathbf{x}}$ to $\mathbf{x}$. The experimental details are given in Appendix C.2.

We compare our methods to three baselines: PCD (Tieleman, 2008), ALOE+ (Dai et al., 2020), and EB-GFN (Zhang et al., 2022a). In Tables 1 and 4, we quantitatively evaluate different methods by evaluating the negative log-likelihood (NLL) and the exponential Hamming MMD (Gretton et al., 2012), respectively. We observe that energy discrepancy outperforms the baseline methods in

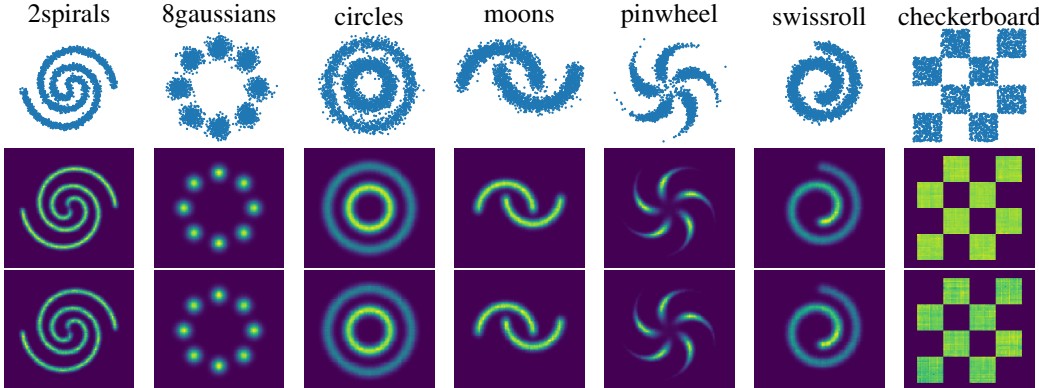

Figure 2: Visualization of training samples and the learned energy landscapes for discrete density estimation. Top to Bottom: training samples, energy landscapes learned by ED-Bern and ED-Grid. We defer more qualitative results to Figure 3.

most settings, all without the need for MCMC simulations (as in PCD) or the training of additional variational networks (like ALOE and EB-GFN). This performance gain is likely explained by the theoretical guarantees of energy discrepancy. In contrast, PCD and ALOE introduce biases due to their reliance on short-run MCMC sampling and variational proposals that may not have converged. Additionally, we provide a qualitative visualisation of the energy landscapes learned by our methods in Figure 2. This visualisation illustrates that energy discrepancy excels at faithfully modeling multi-modal distributions and accurately capturing sharp data support edges. For further qualitative comparisons, please refer to Figure C.2 of Zhang et al. (2022a), which presents energy landscapes for baseline methods.

## 6.3 GRAPH GENERATION

The efficacy of our methods can be further demonstrated by producing high-quality graph samples. Following the setting in You et al. (2018), our model is evaluated on the Ego-small dataset, which comprises one-hop ego graphs extracted from the Citeseer network (Sen et al., 2008). We parametrise the energy function using graph convolutional neural networks (GCNs) (Kipf & Welling, 2016) and train it by minimising the energy discrepancy. After training, new graphs are sampled utilizing the Gibbs-With-Gradient (GWG) sampler (Grathwohl et al., 2021). To assess the quality of these samples, we employ the MMD metric, evaluating it across three graph statistics, *i.e.*, degrees, clustering coefficients, and orbit counts. Additional comprehensive details can be found in Appendix C.3.

Table 2: Graph generation results in terms of MMD. *Avg.* denotes the average over three MMD results.

| Method | Degree | Cluster | Orbit | Avg. |
|---|---|---|---|---|
| GraphVAE | 0.130 | 0.170 | 0.050 | 0.117 |
| DeepGMG | 0.040 | 0.100 | 0.020 | 0.053 |
| GraphRNN | 0.090 | 0.220 | 0.003 | 0.104 |
| GNF | **0.030** | 0.100 | 0.001 | 0.044 |
| GraphAF | **0.030** | 0.110 | **0.001** | 0.047 |
| GraphDF | 0.040 | 0.130 | 0.010 | 0.060 |
| EDP-GNN | 0.052 | 0.093 | 0.007 | 0.050 |
| EBM (GWG) | 0.095 | 0.061 | 0.032 | 0.063 |
| RMwGGIS | 0.066 | 0.042 | 0.036 | 0.048 |
| ED-Bern | 0.063 | 0.054 | 0.014 | 0.044 |
| ED-$\nabla$Bern | 0.033 | 0.046 | 0.020 | **0.033** |
| ED-Grid | 0.036 | 0.050 | 0.019 | 0.035 |
| ED-$\nabla$Grid | 0.040 | **0.040** | 0.021 | 0.036 |

We consider some recent works in graph generation as baselines[4], including GraphVAE (Simonovsky & Komodakis, 2018), DeepGMG (Li et al., 2018), GraphRNN (You et al., 2018), GNF (Liu et al., 2019), GrappAF (Shi et al., 2020), GraphDF (Luo et al., 2021), EDP-GNN (Niu et al., 2020), RMwGGIS (Liu et al., 2023), and contrastive divergence with GWG sampler (Grathwohl et al., 2021). As summarised in Table 2, our methods outperform most baselines in terms of the average of the three MMD metrics, indicating the faithful energy landscapes learned by the energy discrepancy approaches. Additionally, we further visualise the generated samples in Figure 11. It can be seen that the generated samples are one-hop ego graphs, illustrating their adherence to the graph characteristics in the training data.

---

[4]There is insufficient information to reproduce EBM (GwG) and RMwGGIS precisely from Liu et al. (2023). We reran these two baselines with controlled hyperparameters (details are presented in Appendix C.3) for a fair comparison, while other baseline results were taken from their original papers.

Table 3: Experimental results on discrete image modelling. We report the negative log-likelihood (NLL) on the test set for different models. The results of Gibbs, GWG, and DULA are taken from Zhang et al. (2022b), and the result of EB-GFN is from Zhang et al. (2022a).

| Dataset \ Method | Gibbs | GWG | EB-GFN | DULA | ED-Bern | ED-∇Bern | ED-Grid | ED-∇Grid |
|---|---|---|---|---|---|---|---|---|
| Static MNIST | 117.17 | **80.01** | 102.43 | 80.71 | 96.11 | 90.16 | 90.61 | 91.24 |
| Dynamic MNIST | 121.19 | **80.51** | 105.75 | 81.29 | 97.12 | 90.15 | 90.19 | 91.03 |
| Omniglot | 142.06 | 94.72 | 112.59 | 145.68 | 97.57 | 95.56 | **93.94** | 110.31 |

## 6.4 DISCRETE IMAGE MODELLING

Here, we evaluate our methods in discrete high-dimensional spaces. Following the settings in Grathwohl et al. (2021); Zhang et al. (2022b), we conduct experiments on four different binary image datasets. Training details are given in Appendix C.4. After training, we employ annealed importance sampling (Neal, 2001) to estimate the negative log-likelihood (NLL).

The baselines include persistent contrastive divergence with vanilla Gibbs sampling, Gibbs-With-Gradient (Grathwohl et al., 2021, GWG), Generative-Flow-Network (Zhang et al., 2022a, GFN), and Discrete-Unadjusted-Langevin-Algorithm (Zhang et al., 2022b, DULA). Table 3 displays the NLLs on the test dataset. It is evident that energy discrepancy achieves comparable performance to the baseline methods on the Omniglot dataset. Despite the performance gap compared to the contrastive divergence methods on the MNIST dataset, energy discrepancy stands out for its efficiency, requiring only $M$ (in this instance, $M = 32$) evaluations of the energy function per data point, all executed in parallel. This represents a significant computational reduction compared to contrastive divergence, which lacks the advantage of parallelisation and involves simulating multiple MCMC steps. Additionally, our methods show superiority over CD-1 by a substantial margin, as demonstrated in Table 6, affirming the effectiveness of our approach. Notably, ED-∇Bern consistently outperforms ED-Bern, thanks to the efficacy of the gradient-informed proposal. However, ED-∇Grid does not exhibit performance improvement compared to ED-Grid. This is partly because the gradient-informed proposal, when coupled with the grid-neighbourhood transformation, tends to get trapped in local modes as it only flips one bit for each negative sample. For further insights, we provide visualizations of the generated samples in Figure 12, showcasing images generated by our methods.

## 7 CONCLUSION AND OUTLOOK

In this paper we demonstrate that energy discrepancy can be used for efficient and competitive training of energy-based models on discrete data without MCMC. The loss can be defined based on a large class of perturbative processes of which we explore three types, the Bernoulli perturbation (Schröder et al., 2023), the deterministic transform and the neighbourhood-based perturbation. We establish that this simple and computationally cheap approach can achieve competitive results even for intricate data sets such as discrete images.

Furthermore, we introduce a novel interpretation of the negative samples in energy discrepancy via importance sampling. This interpretation allows us to seamlessly incorporate gradient information of the energy function to obtain more informative negative samples while preserving the theoretical guarantees of energy discrepancy. This modification leads to major improvements when learning energy-based models with a Bernoulli perturbation, and drastically outperforms CD-1 despite comparable computational complexity. However, we observe empirically that the gradient information achieves the same improvement for the grid-neighbourhood, despite the success of gradient-informed bit flips in the literature (Grathwohl et al., 2021; Liu et al., 2023). Since the grid-neighbourhood leads to the smallest possible perturbation among all approaches discussed, it is possible that energy discrepancy gets trapped in local modes due to the locality of the approach.

For future work, we are interested in how this work extends to larger highly structured types of data such as molecules or text. These settings may require a deeper understanding of how the perturbation influences the performance of ED and how to harness the information from energy-based models and base distribution in an improved way during training.

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

# Appendix for "Perturb and Learn: Energy-Based Modelling in Discrete Spaces without MCMC"

## CONTENTS

## A  DERIVATION OF THE GRADIENT-INFORMED PROPOSALS

### A.1  GRADIENT-INFORMED PROPOSAL FOR THE BERNOULLI PERTURBATION

In the case of the Bernoulli perturbation, the proposal takes the form

$$\rho_{\mathbf{y}}(\mathbf{x}) = \frac{\prod_{k=1}^{d} \exp\left(-\frac{1}{\tau}\nabla_k^{\mathrm{bit}} U(\mathbf{y})\boldsymbol{\xi}_k\right) \epsilon^{\boldsymbol{\xi}_k}(1-\epsilon)^{(1-\boldsymbol{\xi}_k)}}{Z_{U,\epsilon}} \tag{16}$$

with $\boldsymbol{\xi} = |\mathbf{x} - \mathbf{y}|$, componentwise. To factorise this distribution we need to compute the normalisation $Z_{U,\epsilon}$. This can be done conveniently by iteratively summing over one dimension, i.e. we have

$$Z_{U,\epsilon} = \sum_{\boldsymbol{\xi}\in\{0,1\}^d} \prod_{k=1}^{d} \exp\left(-\frac{1}{\tau}\nabla_k^{\mathrm{bit}} U(\mathbf{y})\boldsymbol{\xi}_k\right) \epsilon^{\boldsymbol{\xi}_k}(1-\epsilon)^{(1-\boldsymbol{\xi}_k)} \tag{17}$$

$$= \sum_{\{\boldsymbol{\xi}_d=0\}} \prod_{k=1}^{d} \exp\left(-\frac{1}{\tau}\nabla_k^{\mathrm{bit}} U(\mathbf{y})\boldsymbol{\xi}_k\right) \epsilon^{\boldsymbol{\xi}_k}(1-\epsilon)^{(1-\boldsymbol{\xi}_k)}$$

$$+ \sum_{\{\boldsymbol{\xi}_d=1\}} \prod_{k=1}^{d} \exp\left(-\frac{1}{\tau}\nabla_k^{\mathrm{bit}} U(\mathbf{y})\boldsymbol{\xi}_k\right) \epsilon^{\boldsymbol{\xi}_k}(1-\epsilon)^{(1-\boldsymbol{\xi}_k)}$$

$$= \left(\exp\left(-\frac{1}{\tau}\nabla_d^{\mathrm{bit}} U(\mathbf{y})\right)\epsilon + 1 - \epsilon\right) \sum_{\boldsymbol{\xi}\in\{0,1\}^{d-1}} \prod_{k=1}^{d-1} \exp\left(-\frac{1}{\tau}\nabla_k^{\mathrm{bit}} U(\mathbf{y})\boldsymbol{\xi}_k\right) \epsilon^{\boldsymbol{\xi}_k}(1-\epsilon)^{(1-\boldsymbol{\xi}_k)}$$

$$= \prod_{k=1}^{d} \left(\exp\left(-\frac{1}{\tau}\nabla_k^{\mathrm{bit}} U(\mathbf{y})\right)\epsilon + 1 - \epsilon\right)$$

Hence, the proposal takes the Bernoulli form

$$\rho_{\mathbf{y}}(\mathbf{x}) = \prod_{k=1}^{d} \left(\frac{\exp\left(-\frac{1}{\tau}\nabla_k^{\mathrm{bit}} U(\mathbf{y})\right)\epsilon}{\exp\left(-\frac{1}{\tau}\nabla_k^{\mathrm{bit}} U(\mathbf{y})\right)\epsilon + 1 - \epsilon}\right)^{\boldsymbol{\xi}_k} \left(\frac{1-\epsilon}{\exp\left(-\frac{1}{\tau}\nabla_k^{\mathrm{bit}} U(\mathbf{y})\right)\epsilon + 1 - \epsilon}\right)^{1-\boldsymbol{\xi}_k} \tag{18}$$

From this, we identify the parameter of the distribution as

$$p_k := \frac{\exp\left(-\frac{1}{\tau}\nabla_k^{\text{bit}}U(\mathbf{y})\right)\epsilon}{\exp\left(-\frac{1}{\tau}\nabla_k^{\text{bit}}U(\mathbf{y})\right)\epsilon + 1 - \epsilon} \tag{19}$$

as claimed.

## A.2  GRADIENT-INFORMED PROPOSAL FOR THE GRID NEIGHBOURHOOD

For the grid neighbourhood, the unnormalised probability is

$$\rho_{\mathbf{y}}(\mathbf{x}) \propto \exp\left(-\frac{1}{\tau}\nabla^{\text{bit}}U(\mathbf{y})^T|\mathbf{x} - \mathbf{y}|\right)\delta_{\mathcal{N}^{-1}(\mathbf{y})}(\mathbf{x}) \tag{20}$$

The inverse grid neighbourhood only consists of elements that differ of $\mathbf{y}$ in exactly one dimension, i.e.

$$\mathcal{N}^{-1}(\mathbf{y}) = \{\mathbf{y}_{\neg k} : k = 1, 2, \ldots, d\} \tag{21}$$

Hence, we can restrict the proposal to this support which allows for tractable normalisation:

$$\rho_{\mathbf{y}}(\mathbf{y}_{\neg k}) = \frac{\exp\left(-\frac{1}{\tau}\nabla_k^{\text{bit}}U(\mathbf{y})\right)}{\sum_{k=1}^d \exp\left(-\frac{1}{\tau}\nabla_k^{\text{bit}}U(\mathbf{y})\right)} = \text{softmax}\left(\exp\left(-\frac{1}{\tau}\nabla^{\text{bit}}U(\mathbf{y})\right)\right)_k \tag{22}$$

# B  DETERMINISTIC TRANSFORMATION AND PROOF OF CONSISTENT APPROXIMATION

## B.1  MEAN POOLING TRANSFORM

We describe the mean-pooling transform on the example of image data which takes values in the space $\{0,1\}^{h \times w}$. We fix a window size $s$ and reshape each data-point into blocks of size $s \times s$, i.e.

$$\{0,1\}^{h \times w} \rightarrow \{0,1\}^{s \times s \times \frac{h}{s} \times \frac{w}{s}}, \quad \mathbf{x} \mapsto \bar{\mathbf{x}}$$

The mean pooling transform $g_{\text{pool}}$ computes the average over each block $\bar{\mathbf{x}}_{\bullet,\bullet,i,j}$ for $i = 1, 2, \ldots, h/s$ and $j = 1, 2, \ldots, w/s$. The corresponding preimage of the mean pooling transform is given by the set of points which are identical to $\mathbf{x}$ up to block-wise permutation, i.e.

$$g^{-1}(g_{\text{pool}}(\mathbf{x})) = \{\mathbf{x}' \in \mathcal{X} : \text{ there exist } \pi_{i,j} \in S_{s \times s} \text{ s.t. } \bar{\mathbf{x}}'_{l,k,i,j} = \bar{\mathbf{x}}'_{\pi_{i,j}(l,k),i,j} \text{ for all } l, k, i, j\}$$

where $S_{s \times s}$ denotes the permutation group for matrices of size $s \times s$. In practice, the mean-pooled data point has to never be computed, only the block wise permutations of the data point are required. Consequently, we obtain negative samples through $\mathbf{x}_-^{i,j} \sim \mathcal{U}(g^{-1}(g_{\text{pool}}(\mathbf{x}^i)))$, i.e. via block wise permutation of the entries of each data point $\mathbf{x}^i$.

Strictly speaking, this transformation violates the assumptions of Proposition 1 for data points that only consist of blocks that average to 1 or 0. Since this is only the case for a small set of the state space, we assume this violation to be negligible.

## B.2  CONSISTENCY OF OUR APPROXIMATION

The following proof is similar to Schröder et al. (2023). We first restate the consistency result:

**Theorem 1.** *For every $\varepsilon > 0$ there exist $N, M \in \mathbb{N}$ such that $|\mathcal{L}_{q,M,w}(U) - \text{ED}_q(p_{\text{data}}, U)| < \varepsilon$ a.s..*

*Proof.* For $N$ data points $\mathbf{x}_+^i \sim p_{\text{data}}$ and perturbed points $\mathbf{y}^i \sim q(\cdot|\mathbf{x}_+^i)$, denote the $M$ corresponding negative samples by $\mathbf{x}_-^{i,j} \sim \rho_{\mathbf{y}^i}(\mathbf{x})$, where $\rho_{\mathbf{y}^i}(\mathbf{x})$ can be the uninformed or gradient-informed proposal. Notice that the distribution of the negative samples depends on $\mathbf{y}^i$. Using the triangle inequality, we can upper bound the difference $|\text{ED}_q(p_{\text{data}}, U) - \mathcal{L}_{q,M,w}(U)|$ by upper bounding the

Table 4: Experimental results of discrete density estimation. We display the MMD (in units of $1 \times 10^{-4}$). The results of baselines are taken from Zhang et al. (2022a).

| Metric | Method | 2spirals | 8gaussians | circles | moons | pinwheel | swissroll | checkerboard |
|--------|--------|----------|-----------|---------|-------|----------|-----------|--------------|
| MMD↓ | PCD | 2.160 | 0.954 | 0.188 | 0.962 | 0.505 | 1.382 | 2.831 |
| | ALOE+ | 0.149 | 0.078 | 0.636 | 0.516 | 1.746 | 0.718 | 12.138 |
| | EB-GFN | 0.583 | 0.531 | 0.305 | 0.121 | 0.492 | 0.274 | **1.206** |
| | ED-Bern | 0.120 | 0.014 | 0.137 | **-0.088** | 0.046 | 0.045 | 1.541 |
| | ED-∇Bern | 0.148 | -0.059 | **-0.028** | 0.060 | **0.045** | **0.015** | 1.276 |
| | ED-Grid | **0.097** | **-0.066** | 0.022 | 0.018 | 0.351 | 0.097 | 2.049 |
| | ED-∇Bern | 2.079 | 0.912 | 0.185 | 1.397 | 0.535 | 0.235 | 6.901 |

following two terms, individually:

$$
\left| \mathrm{ED}_q(p_{\mathrm{data}}, U) - \frac{1}{N} \sum_{i=1}^{N} \log \mathbb{E} \left[ \exp(U(\mathbf{x}_+^i) - U(\mathbf{x}_-^{i,j}) + \log w_{\mathbf{y}^i}(\mathbf{x}_-^{i,j})) \, \middle| \, \mathbf{x}_+^i, \mathbf{y}^i \right] \right|
$$

$$
+ \left| \frac{1}{N} \sum_{i=1}^{N} \log \mathbb{E} \left[ \exp(U(\mathbf{x}_+^i) - U(\mathbf{x}_-^{i,j}) + \log w_{\mathbf{y}^i}(\mathbf{x}_-^{i,j})) \, \middle| \, \mathbf{x}_+^i, \mathbf{y}^i \right] - \mathcal{L}_{q,M,w}(U) \right|
$$

The conditioning expresses that the expectation is only taken in $\mathbf{x}_-^{i,j} \sim \rho_{\mathbf{y}^i}(\mathbf{x})$ while keeping the values of the random variables $\mathbf{x}_+^i$ and $\mathbf{y}^i$ fixed. The first term can be bounded by a sequence $\varepsilon_N \xrightarrow{a.s.} 0$ due to the normal strong law of large numbers. For the second term one needs to consider that the distribution $\rho_{\mathbf{y}^i}(\mathbf{x})$ depends on the random variable $\mathbf{y}^i$. For this reason, we notice that $\mathbf{x}_-^{i,j}$ are conditionally independent given $\mathbf{x}_+^i, \mathbf{y}^i$ and employ a conditional version of the strong law of large numbers (Majerek et al., 2005, Theorem 4.2) to obtain

$$
\frac{1}{M} \sum_{j=1}^{M} \exp \left( U(\mathbf{x}_+^i) - U(\mathbf{x}_-^{i,j}) + \log w_{\mathbf{y}^i}(\mathbf{x}_-^{i,j}) \right) \xrightarrow{a.s.} \mathbb{E} \left[ \exp(U(\mathbf{x}_+^i) - U(\mathbf{x}_-^{i,j}) + \log w_{\mathbf{y}^i}(\mathbf{x}_-^{i,j}) \, \middle| \, \mathbf{x}_+^i, \mathbf{y}^i \right]
$$

Next, we have that the deterministic sequence $w/M \to 0$. Thus, adding the stabilisation $w/M$ does not change the limit in $M$. Furthermore, since the logarithm is continuous, the limit also holds after applying the logarithm. Finally, the estimate translates to the sum by another application of the triangle inequality: For each $i = 1, 2, \ldots, N$ there exists a sequence $\varepsilon_{i,M} \xrightarrow{a.s.} 0$ such that

$$
\left| \frac{1}{N} \sum_{i=1}^{N} \log \mathbb{E} \left[ \exp(U(\mathbf{x}_+^i) - U(\mathbf{x}_-^{i,j}) + \log w_{\mathbf{y}^i}(\mathbf{x}_-^{i,j}) \, \middle| \, \mathbf{x}_+^i, \mathbf{y}^i \right] - \mathcal{L}_{q,M,w}(U) \right|
$$

$$
\leq \frac{1}{N} \sum_{i=1}^{N} \left| \log \mathbb{E} \left[ \exp(U(\mathbf{x}_+^i) - U(\mathbf{x}_-^{i,j}) + \log w_{\mathbf{y}^i}(\mathbf{x}_-^{i,j}) \, \middle| \, \mathbf{x}_+^i, \mathbf{y}^i \right] - \mathcal{L}_{q,M,w}(U) \right|
$$

$$
< \frac{1}{N} \sum_{i=1}^{N} \varepsilon_{i,M} \leq \max(\varepsilon_{1,M}, \ldots, \varepsilon_{N,M}).
$$

Hence, for each $\varepsilon > 0$ there exists an $N \in \mathbb{N}$ and an $M(N) \in \mathbb{N}$ such that $|\mathrm{ED}_q(p_{\mathrm{data}}, U) - \mathcal{L}_{q,M(N),w}(U)| < \varepsilon$ almost surely. □

## C  MORE ABOUT EXPERIMENTS

### C.1  TRAINING ISING MODELS

**Experimental Details.** As in Grathwohl et al. (2021); Zhang et al. (2022a;b), we train a learnable connectivity matrix $J_\phi$ to estimate the true matrix $J$ in the Ising model. To generate the training data, we simulate Gibbs sampling with $1,000,000$ steps for each instance to construct a dataset of $2,000$ samples. For energy discrepancy, we choose $w = 1, M = 32$ for all variants, $\epsilon = 0.1$ in

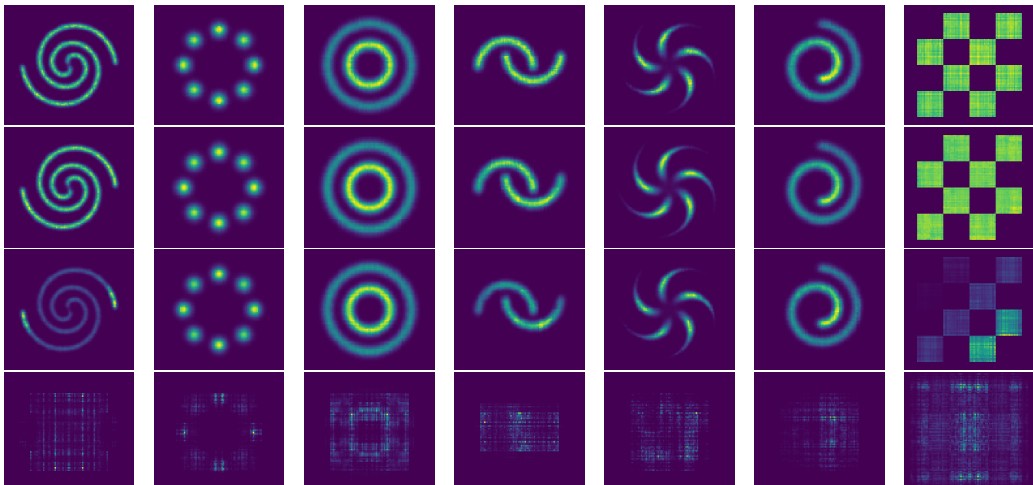

Figure 3: Visualization of the learned energy landscapes. Top to bottom: ED-Pool, ED-∇Bern, ED-∇Grid, ED-Gaussian. The last row illustrates the failure of employing energy discrepancy with Gaussian perturbation in discrete spaces.

Bernoulli perturbations, and $\tau = 2$ in gradient-informed proposals. The parameter $J_\phi$ is learned by the Adam (Kingma & Ba, 2014) optimizer with a learning rate of $0.0001$ and a batch size of 256. Following Zhang et al. (2022a), all models are trained with an $l_1$ regularization with a coefficient in $\{100, 50, 10, 5, 1, 0.1, 0.01\}$ to encourage sparsity. The other setting is basically the same as Section F.2 in Grathwohl et al. (2021). We report the best result for each setting using the same hyperparameter searching protocol for all methods.

**Quantitative Results.** We consider $D = 10 \times 10$ grids with $\sigma = 0.1, 0.2, \ldots, 0.5$ and $D = 9 \times 9$ grids with $\sigma = -0.1, -0.2$. The methods are evaluated by computing the negative log-RMSE between the estimated $J_\phi$ and the ture matrix $J$. As shown in Table 5, our methods demonstrate comparable results to the baselines and, in certain settings, even outperform Gibbs and GWG, indicating that energy discrepancy is able to discover the underlying structure within the data.

Table 5: Mean negative log-RMSE (higher is better) between the learned connectivity matrix $J_\phi$ and the true matrix $J$ for different values of $D$ and $\sigma$. The results of baselines are directly taken from Zhang et al. (2022a).

| Method \ $\sigma$ | $D = 10^2$ | | | | | $D = 9^2$ | |
|---|---|---|---|---|---|---|---|
| | 0.1 | 0.2 | 0.3 | 0.4 | 0.5 | −0.1 | −0.2 |
| Gibbs | 4.8 | 4.7 | **3.4** | **2.6** | **2.3** | 4.8 | 4.7 |
| GWG | 4.8 | 4.7 | **3.4** | **2.6** | **2.3** | 4.8 | 4.7 |
| EB-GFN | **6.1** | **5.1** | 3.3 | **2.6** | **2.3** | **5.7** | **5.1** |
| ED-Bern | 5.1 | 4.0 | 2.9 | 2.5 | **2.3** | 5.1 | 4.3 |
| ED-∇Bern | 5.0 | 4.2 | 3.0 | **2.6** | **2.3** | 5.0 | 4.2 |
| ED-Grid | 4.6 | 4.0 | 3.1 | **2.6** | **2.3** | 4.5 | 4.0 |
| ED-∇Grid | 4.6 | 4.1 | 3.2 | **2.6** | **2.3** | 4.8 | 4.7 |

## C.2 Discrete Density Estimation

**Experimental Details.** This experiment keeps a consistent setting with Dai et al. (2020). We first generate 2D floating-points from a continuous distribution $\hat{p}$ which lacks a closed form but can be easily sampled. Then, each sample $\hat{\mathbf{x}} := [\hat{\mathbf{x}}_1, \hat{\mathbf{x}}_2] \in \mathbb{R}^2$ is converted to a discrete data point $\mathbf{x} \in \{0, 1\}^{32}$ using Gray code. To be specific, given $\hat{\mathbf{x}} \sim \hat{p}$, we quantise both $\hat{\mathbf{x}}_1$ and $\hat{\mathbf{x}}_2$ into 16-bits binary representations via Gray code (Gray, 1953), and concatenate them together to obtain a 32-bits vector $\mathbf{x}$. As a result, the probabilistic mass function in the discrete space is $p(\mathbf{x}) \propto \hat{p}([\text{GrayToFloat}(\mathbf{x}_{1:16}), \text{GrayToFloat}(\mathbf{x}_{17:32})])$. It is noteworthy that learning on this discrete space presents challenges due to the highly non-linear nature of the Gray code transformation.

The energy function is parameterised by a 4 layer MLP with 256 hidden dimensions and Swish (Ramachandran et al., 2017) activation. We train the EBM for $10^5$ steps and adopt an Adam optimiser with a learning rate of $0.002$ and a batch size of 128 to update the parameter. For the energy discrepancy, we choose $w = 1, M = 32$ for all variants, $\epsilon = 0.1$ in Bernoulli perturbations, and

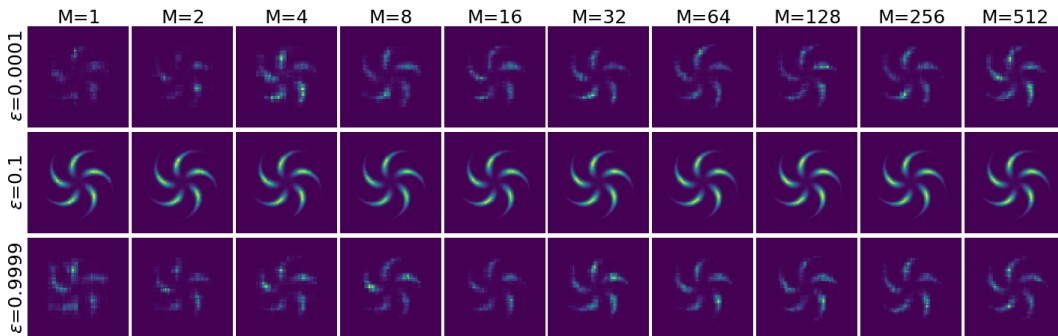

Figure 5: Density estimation results of ED-Bern on the pinwheel with different $\epsilon$, $M$ and $w = 1$.

$\tau = 2$ in gradient-informed proposals. For deterministic transformation, we use mean pooling with the window size $32 \times 1$ in EDP. After training, we quantitatively evaluate all methods using the negative log-likelihood (NLL) and the maximum mean discrepancy (MMD). To be specific, the NLL metric is computed based on $4,000$ samples drawn from the data distribution, and the normalisation constant is estimated using importance sampling with $1,000,000$ samples drawn from a variational Bernoulli distribution with $p = 0.5$. For the MMD metric, we follow the setting in Zhang et al. (2022a), which adopts the exponential Hamming kernel with $0.1$ bandwidth. Moreover, the reported performances are averaged over 10 repeated estimations, each with $4,000$ samples, which are drawn from the learned energy function via Gibbs sampling.

**The Effect of $\epsilon$ in Bernoulli Perturbation.** Perhaps surprisingly, we find that the proposed energy discrepancy loss with Bernoulli perturbation is very robust to the noise scalar $\epsilon$. In Figure 4, we visualise the learned energy landscapes with different $\epsilon$. The results demonstrate that ED-Bern is able to learn faithful energy functions, even with extreme values of $\epsilon$, such as $\epsilon \in \{0.999, 0.001\}$. This highlights the robustness and effectiveness of our approach. In Figure 5,

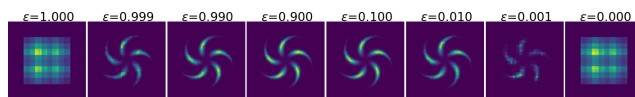

Figure 4: Density estimation results of ED-Bern on the pinwheel with different $\epsilon$ and $M = 32, w = 1$.

we further show that, with $\epsilon \in \{0.9999, 0.0001\}$, ED-Bern can still learn a faithful energy landscape using a large value of $M$. However, when $\epsilon \in \{1, 0\}$, ED-Bern fails to work. It is noteworthy that the choice of $\epsilon$ is highly dependent on the specific structure of the dataset. While ED-Bern exhibits robustness to different values of $\epsilon$ in the synthetic data, we have observed that a large value of $\epsilon$ ($\epsilon \geq 0.1$) is not effective for discrete image modelling.

**The Effect of Window Size in Mean Pooling Transformation.** To investigate the effectiveness of the window size in ED-Pool, we conduct experiments in Figure 6 with different window sizes. The results indicate that employing a small window size (e.g., $2 \times 1$) does not provide sufficient information for energy discrepancy to learn the underlying data structure effectively. Furthermore, our empirical findings suggest that solely increasing the value of $M$ is not a viable solution to address this issue. Again, the choice of the window size should depend on the underlying data structure. In the discrete image modelling, we find that even with a small window size (i.e., $4 \times 4$), energy discrepancy yields energy with

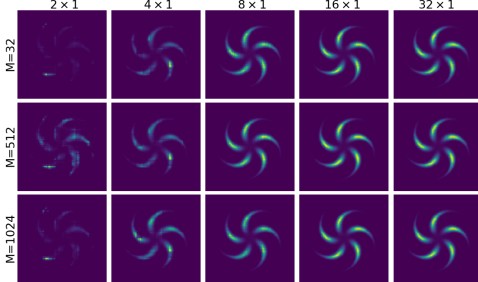

Figure 6: Density estimation results of ED-Pool on the pinwheel with different window sizes, $M$ and $w = 1$.

low values on the data-support but rapidly diverging values outside of it. Therefore, it fails to learn a faithful energy landscape.

**Qualitatively Understanding the Effect of $w$ and $M$.** The hyperparameters $w$ and $M$ play a crucial role in the estimation of energy discrepancy. Increasing $M$ can reduce the variance of the Monte Carlo estimation of the contrastive potential in (3), while a proper value of $w$ can improve the stabilisation of training. Here, we evaluate the effect of $w$ and $M$ on the variants of energy discrepancy in Figures 7

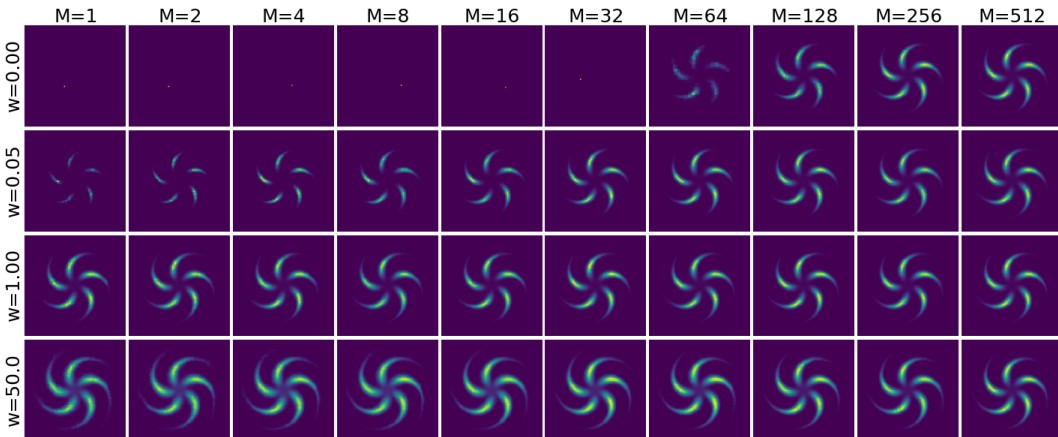

Figure 7: Density estimation results of ED-Bern on the pinwheel with different $w, M$ and $\epsilon = 0.1$.

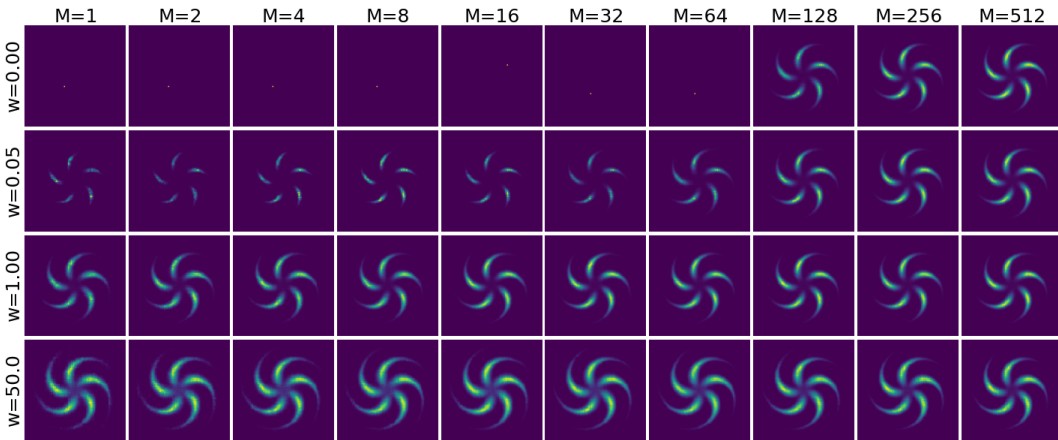

Figure 8: Density estimation results of ED-Pool on the pinwheel with different $w, M$ and the window size is $32 \times 1$.

to 9. Based on empirical observations, we observe that when $w = 0$ and $M$ is small (*e.g.*, $M \leq 32$ for ED-Bern and $M \leq 64$ for ED-Pool and ED-Grid), energy discrepancy demonstrates rapid divergence and fails to converge. Additionally, we find that increasing $M$ can address this issue to some extent and introducing a non-zero value for $w$ can significantly stabilize the convergence, even with $M = 1$. Moreover, larger $w$ tends to produce a flatter estimated energy landscapes, which also aligns with the findings in continuous scenarios of energy discrepancy (Schröder et al., 2023).

**The Failure of Gaussian Perturbation.** As highlighted in Grathwohl et al. (2021), the deep energy $U_\theta(\mathbf{x})$ is a differentiable function that accepts real-valued inputs, despite being evaluated solely on a discrete subset of their domain. This observation inspires us to train discrete EBMs using continuous relaxation. Consequently, we train the energy function $U_\theta$ using Gaussian perturbations as proposed in (Schröder et al., 2023, see the loss function $\mathcal{L}_{t,M,w}(\theta)$). Specifically, we chose $w = 1, M = 32$ and the noise scale at $t = 0.5$, which is the optimal choice among the range of $\{0.01, 0.1, 0.5, 1\}$. We demonstrate the learned energy landscape in the last row of Figure 3. Notably, it is evident that the energy discrepancy with Gaussian perturbation falls short when it comes to training energy-based models on discrete data. This inadequacy prompts the need for the development of new perturbations tailored to the challenges posed by discrete datasets.

### C.3 GRAPH GENERATION

**Experimental Details.** In this experiment, we assess the performance of our proposed methods using the Ego-small dataset, which consists of 200 ego graphs. Following the setup in You et al. (2018), 80% of these graphs are allocated for training, with the remaining 20% designated for testing. To

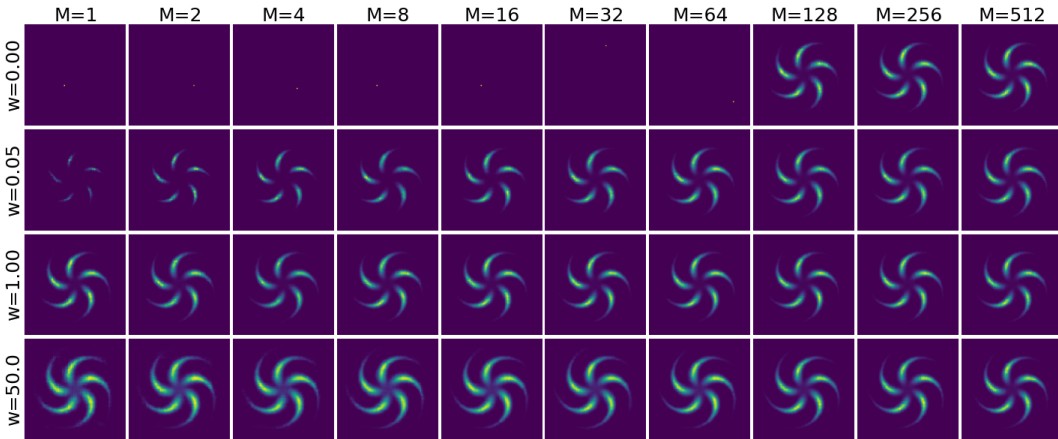

Figure 9: Density estimation results of ED-Grid on the pinwheel with different $w, M$.

provide better insight into this task, we illustrate a subset of training data in Figure 10. Notably, these training data examples closely resemble realistic one-hop ego graphs.

For a fair comparison, we parametrise the energy function via a 5-layer GCN (Kipf & Welling, 2016) with the ReLU activation and 16 hidden states for all energy-based approaches, {i.e.,} EBM (GWG), RMwGGIS, and the variants of energy discrepancy. For hyperparameters, we choose $M = 32, w = 1$ for all variants of energy discrepancy and $\epsilon = 0.1$ for ED-Bern and ED-$\nabla$Bern. Following the configuration in Liu et al. (2023), we apply the advanced version of RMwGGIS with the number of samples $s = 50$ (Liu et al., 2023, Equation 11). Regarding the EBM (GWG) baseline, we train it using persistent contrastive divergence with a buffer size of 200 samples and the MCMC steps being 50.

To train the models, we use the Adam optimizer with a learning rate of 0.0001 and a batch size of 200. After training, we generate new graphs by first sampling $N$, which is the number of nodes to be generated, from the empirical distribution of the number of nodes in the training dataset, and then applying the GWG sampler with 50 MCMC steps from a randomly initialised Bernoulli noise. Following the evaluation scheme in Liu et al. (2019), We trained 5 separate models of each type and performed 3 trials per model, then averaged the result over 15 runs.

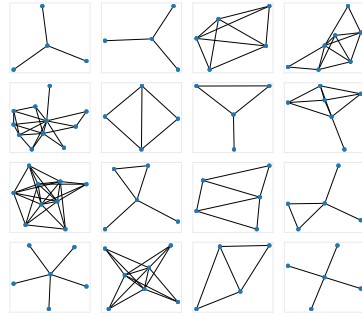

**Qualitative Results.** We provide a visualization of generated graphs from variants of our methods in Figure 11. Notably, the majority of these generated graphs resemble one-hop ego graphs, demonstrating the faithfulness of the energy landscapes learned through energy discrepancies.

Figure 10: Illustration of the training data in graph generation.

## C.4 DISCRETE IMAGE MODELLING

**Experimental Details.** In this experiment, we parametrise the energy function using ResNet (He et al., 2016) following the settings in Grathwohl et al. (2021); Zhang et al. (2022b), where the network has 8 residual blocks with 64 feature maps. Each residual block has 2 convolutional layers and uses Swish activation function (Ramachandran et al., 2017). We choose $M = 32, w = 1$ for all variants of energy discrepancy, $\epsilon = 0.001$ in Bernoulli perturbations, and $\tau = 2$ in gradient-informed proposals. Note that here we choose a relatively small $\epsilon$ since we empirically find that the loss of energy discrepancy converges to a constant rapidly with larger $\epsilon$, which can not provide meaningful gradient information to update the parameters. All models are trained with Adam optimiser with a learning rate of 0.0001 and a batch size of 100 for $50,000$ iterations. We perform model evaluation every $5,000$ iteration by conducting Annealed Importance Sampling (AIS) with a Gibbs-With-Gradient sampler for $10,000$ steps. The reported results are obtained from the model that achieves the best performance on the validation set. After training, we finally report the negative log-likelihood by running $300,000$ iterations of AIS.

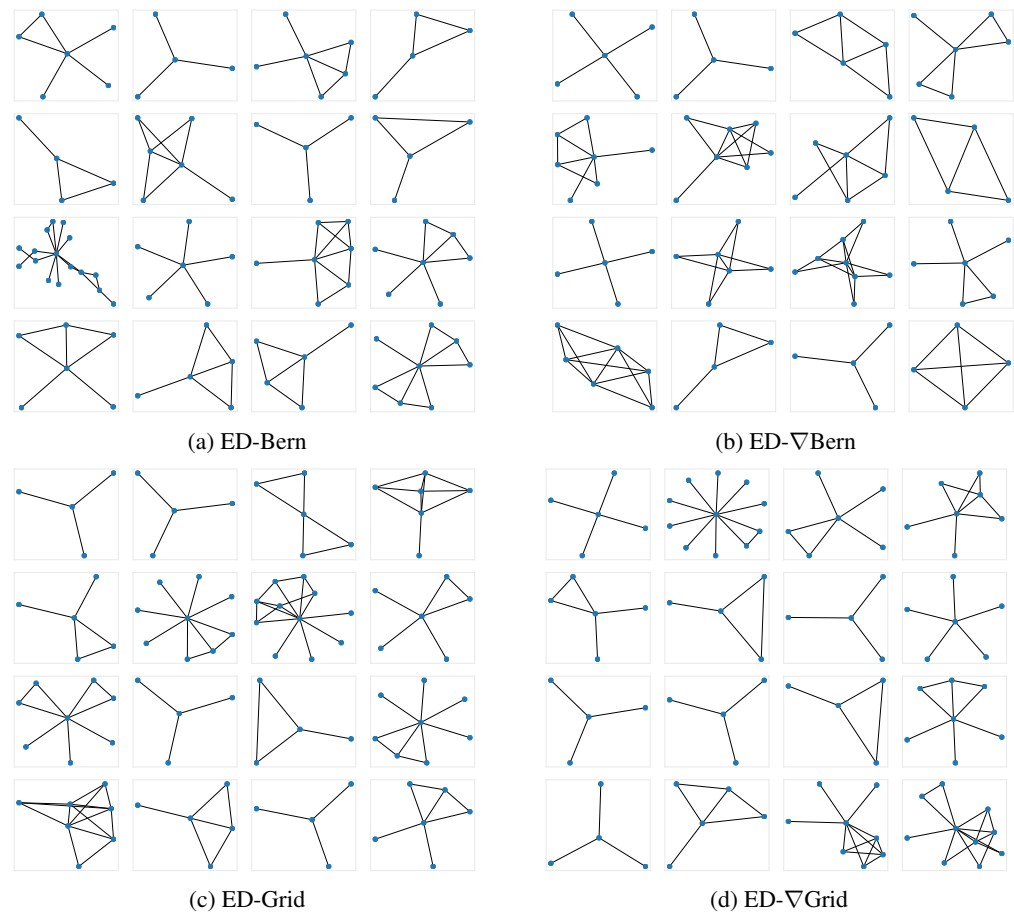

(a) ED-Bern

(b) ED-∇Bern

(c) ED-Grid

(d) ED-∇Grid

Figure 11: Visualisation of the samples drawn from the energy-based models learned by the variants of our approaches on the Ego-small dataset.

**Qualitative Results.** The images generated in the final step of AIS are displayed in Figure 12. Although these samples exhibit favourable likelihoods, it is apparent that they still contend with mode collapse. We posit that this challenge could potentially be mitigated by employing more advanced sampling techniques, which we leave for future research.

**Comparison to Contrastive Divergence with Different MCMC Steps.** Given the resemblance between energy discrepancy with gradient-informed proposals and contrastive divergence, this study undertakes a thorough comparison of these two methods. Specifically, we employ the officially open-sourced implementation[5] of DULA to conduct contrastive divergence training, with varying MCMC steps. As depicted in Table 6, our findings indicate that energy discrepancy significantly outperforms contrastive divergence when employing a single MCMC step, and achieves performance comparable to CD-10. We attribute this superiority to the fact that CD-1 involves a biased estimation of the log-likelihood gradient due to inherent issues with non-convergent MCMC processes. In contrast, energy discrepancy mitigates this problem by introducing important weights, resulting in a consistent approximation, as indicated in Theorem 1.

**Time Complexity Comparison for Energy Discrepancy and Contrastive Divergence.** In this experiment, we evaluate the running time per iteration and epoch for energy discrepancy and contrastive divergence in training a discrete EBM on the static MNIST dataset. The experiments include contrastive divergence with varying MCMC steps and variants of energy discrepancy with a fixed value of $M = 4$. The results, presented in Table 7, highlight that ED-Bern and ED-Grid are the fastest options, as they don't involve gradient computations during training. In contrast, ED-∇Bern and

---

[5]https://github.com/ruqizhang/discrete-langevin

Table 6: Experimental results of the comparison between energy discrepancy and contrastive divergence with varying MCMC steps.

| Dataset \ Method | CD-1 | CD-3 | CD-5 | CD-7 | CD-10 | ED-Bern | ED-$\nabla$Bern | ED-Grid | ED-$\nabla$Grid |
|---|---|---|---|---|---|---|---|---|---|
| Static MNIST | 182.53 | 130.94 | 102.70 | 98.07 | **88**.13 | 96.11 | 90.16 | 90.61 | 91.24 |
| Dynamic MNIST | 157.14 | 130.56 | 97.50 | 91.00 | **84**.16 | 97.12 | 90.15 | 90.19 | 91.03 |
| Omniglot | nan. | 161.96 | 142.91 | 149.68 | 146.11 | 97.57 | 95.56 | **93**.94 | 110.31 |

Table 7: Running time complexity comparison for energy discrepancy and contrastive divergence.

| Time \ Method | CD-1 | CD-5 | CD-10 | ED-Bern | ED-$\nabla$Bern | ED-Grid | ED-$\nabla$Grid |
|---|---|---|---|---|---|---|---|
| Per Iteration (s) | 0.0583 | 0.1904 | 0.3351 | 0.0490 | 0.0576 | 0.0479 | 0.0630 |
| Per Epoch (s) | 29.1660 | 95.2178 | 167.5718 | 24.5265 | 28.8305 | 23.9861 | 31.5008 |

ED-$\nabla$Grid require one gradient computation for each parameter update, placing them in a comparable computational complexity range with CD-1 and expected to be more efficient than CD-10.

**The Efficacy of the Number of Negative Samples.** In all experiments, we selected the number of negative samples as $M = 32$ irrespective of the dimension of the problem, to maximise computational efficiency within the constraints of our GPU capacity. We have demonstrated in Figures 7 to 9 that our results remain largely unchanged when choosing the number of negative samples as smaller or larger.

To further investigate, we conduct additional experiments by training energy-based models on the static MNIST dataset with ED-Grid for different values of $M$. As detailed in Table 8, our results maintain comparable quality even as the number of negative samples is decreased. Notably, our approach offers greater parallelization potential compared to the sequentially computed MCMC of contrastive divergence.

Table 8: Discrete image modelling results of ED-Grid on the static MNIST dataset with different $M$ and $w = 1$.

| | $M = 4$ | $M = 8$ | $M = 16$ | $M = 32$ |
|---|---|---|---|---|
| NLL | 90.13 | 90.37 | 89.14 | 90.61 |

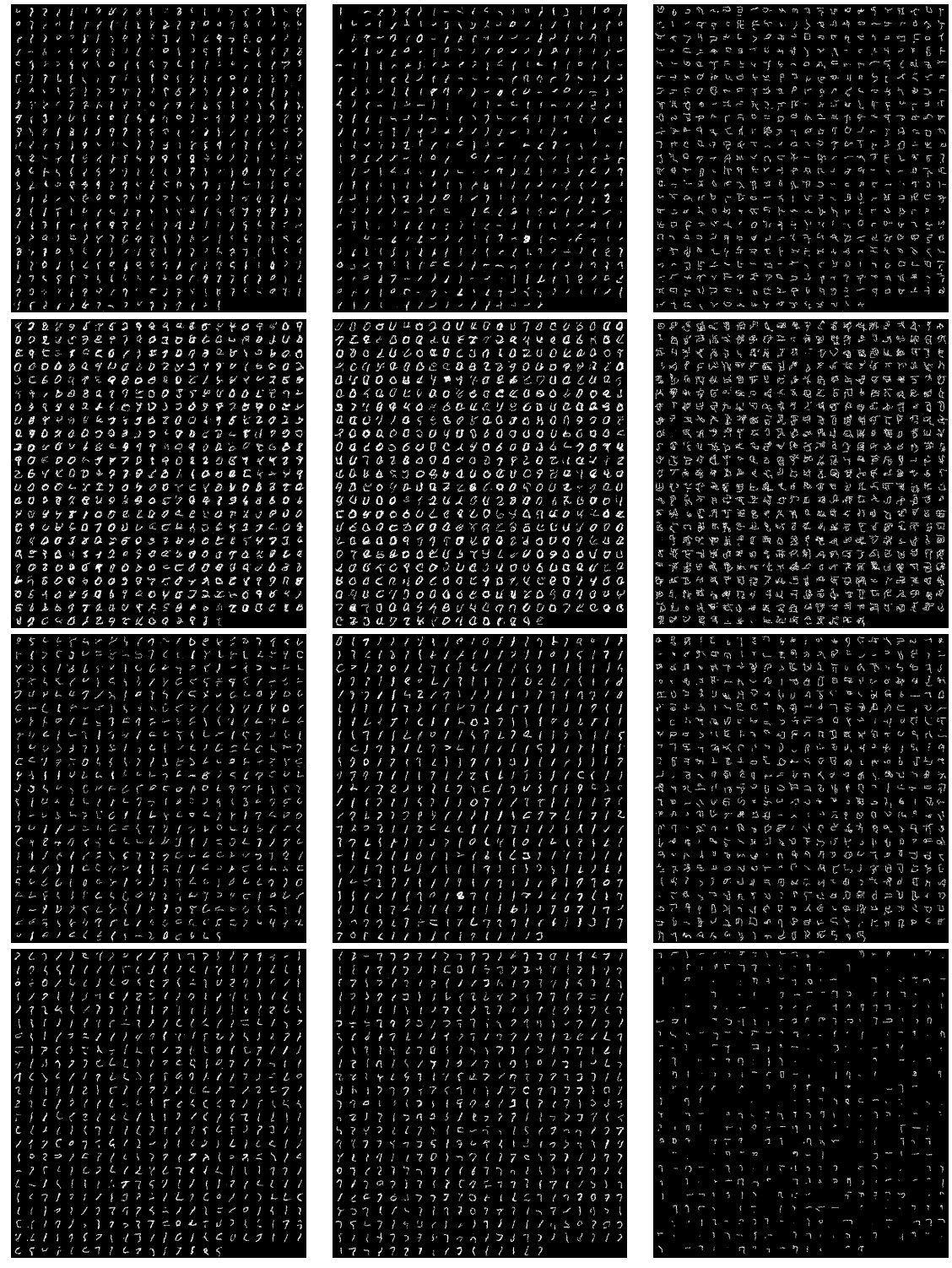

Figure 12: Generated samples on discrete image modelling. Left to right: Static MNIST, Dynamic MNIST, Omniglot. Top to bottom: ED-Bern, ED-∇Bern, ED-Grid, ED-∇Grid.

