# OpenReview forum: "Perturb and Learn: Energy-Based Modelling in Discrete Spaces without MCMC"
_ICLR.cc/2024/Conference — Submitted to ICLR 2024_

### Official Review · Reviewer_kTVF · 2023-10-23

**Soundness:** 4 excellent
**Presentation:** 4 excellent
**Contribution:** 4 excellent
**Rating:** 8
**Confidence:** 4

**Summary:**

This paper proposes training discrete EBMs using Energy Discrepancy (ED) instead of Contrastive Divergence (CD). Training with ED does not require sampling, instead it only requires evaluating the energy function at data points and their perturbations.

The paper claims that existing approaches suffer from a tradeoff between obtaining more accurate samples during training, and increasing the time-cost of training, and obtaining less accurate samples, increasing the bias of the gradients used during training.

To apply this technique, one must choose a suitable distribution to sample data perturbations, and an efficient method to estimate the contrastive potential induced by this distribution on data perturbations.

The authors propose several data perturbation distributions, and propose estimating the contrastive potential using importance sampling. For the proposal distribution, the authors investigate using uninformed proposals, and gradient-informed proposals. Uninformed proposals do not exploit any information about the learned energy function U.

Gradient-informed proposals used a Taylor expansion similar to that shown in Gibbs-with-Gradients (GwG). The authors draw a connection between their approach and GwG.

Finally, the authors train a lower bound on this loss to improve training stability. The authors present several configurations of their approach and compare to baselines.

The authors present results on small toy problems for visualisation, as well as on graph generation and image generation.

The authors present thorough ablations and experimental details in the appendix.

**Strengths:**

Very thorough ablations are presented in the appendix, investigating the effects of a number of parameters to configure their method.

**Weaknesses:**

The results on discrete image modelling are not very strong, either in terms of sample quality or in terms of estimated NLL.

**Questions:**

I’m curious about the wall-clock speed of training. Your approach admits better parallelisation than existing techniques since it relies on importance sampling rather than MCMC. Do you have any wall-clock timing experiments investigating this? For example, how does the wall-clock time compare of your approach vs. CD-10?

Must the GwG sampler be tuned differently when training with ED vs. GwG? i.e. do you find that sampling is less efficient at test-time when a model is trained with ED?

---

> ### Author Response · Authors · 2023-11-19
>
> Thanks for your positive review of our work. Here are our answers to your questions and comments.
>
> > The results on discrete image modelling are not very strong, either in terms of sample quality or in terms of estimated NLL.
> >
>
> **ANSWER:** Thanks for pointing this out. Currently, there is still a performance gap of Energy discrepancy to MCMC-based discrete EBM training methods like Contrastive Divergence with ten steps of GWG [1], or DULA [2] on discrete image modelling tasks. There are two promising directions for future work to improve our results: refining the perturbation $q(y|x)$ in image modelling and improving the estimation of the energy discrepancy loss with better importance sampling distributions.
>
> However, our methods do outperform CD variants that use smaller numbers of MCMC steps. Thus, we see our key contribution in demonstrating that by employing simple perturbations such as Bernoulli, energy discrepancy can effectively learn various types of data distributions explored in previous discrete EBM literature with a lower computational cost than CD-1.
>
> > I’m curious about the wall-clock speed of training. Your approach admits better parallelisation than existing techniques since it relies on importance sampling rather than MCMC. Do you have any wall-clock timing experiments investigating this? For example, how does the wall-clock time compare of your approach vs. CD-10?
> >
>
> **ANSWER:** Great suggestions! ED-Bern and ED-Grid don’t rely on any gradient computation during training, making them the fastest options. ED-$\nabla$Bern and ED-$\nabla$Grid necessitate one gradient computation for each parameter update. As a result, they share a comparable computational complexity with CD-1 and are expected to be more efficient than CD-10. Here, we present a comparison of average timings for 1 batch/epoch of CD-1, CD-5, and CD-10 as well as four variants of our methods using $M=4$ contrastive samples (training is conducted on static MNIST with a batch size of $100$). We further report the NLL of contrastive divergence and ED-Grid. One can see that a comparable accuracy to contrastive divergence can be achieved at a fraction of the computation time. (The NLL of ED-Bern and the gradient guided methods was reported in the main paper but were computed for $M=32$ contrastive samples.)
>
> |  | CD-1 | CD-5 | CD-10 | ED-Bern | ED-$\nabla$Bern | ED-Grid | ED-$\nabla$Grid |
> | --- | --- | --- | --- | --- | --- | --- | --- |
> | per update (s) | 0.0583 | 0.1904 | 0.3351 | 0.0490 | 0.0576 | 0.0479 | 0.0630 |
> | per epoch (s) | 29.1660 | 95.2178 | 167.5718 | 24.5265 | 28.8305 | 23.9861 | 31.5008 |
> | NLL | 182.53 | 102.70 | 88.13 | - | - | 90.13 | - |
>
> > Must the GwG sampler be tuned differently when training with ED vs. GwG? i.e. do you find that sampling is less efficient at test-time when a model is trained with ED?
> >
>
> **ANSWER:** This is indeed an interesting question. In our experiments, the negative log-likelihoods were computed with annealed importance sampling, using the same tuning as in Gibbs with Gradients [1]. In graph generation, the GwG sampler was also utilised without any additional tuning, consistently producing high-quality graph structures. In addition, sampling from energies learned with ED is not less efficient than sampling from models learned with CD.
>
> This being said, it was shown in [3] that energy-based models obtained from short-run MCMC should also use short-run MCMC for sample synthesis. Models learned with energy discrepancy, on the other hand, have not been learned with short-run MCMC and should benefit more from a long-run Markov chain when synthesising new samples. Furthermore, it is an interesting question if the choice of perturbation also suggests a certain choice of sampler, e.g. if EBMs learned with the grid neighbourhood perturbation should be sampled from using GwG. As of now, we don’t have a complete picture answering these questions because assessing sample quality is non-trivial.
>
> [1] Grathwohl, Will, et al. "Oops i took a gradient: Scalable sampling for discrete distributions." *International Conference on Machine Learning*. PMLR, 2021.
>
> [2] Zhang, Ruqi, Xingchao Liu, and Qiang Liu. "A Langevin-like sampler for discrete distributions." *International Conference on Machine Learning*. PMLR, 2022.
>
> [3] Nijkamp, Erik et al. "On the Anatomy of MCMC-based Maximum Likelihood Learning of Energy-Based Models", AAAI, 2019

---

> > ### Comment · Reviewer_kTVF · 2023-11-22
> > **Response to Authors Comments**
> >
> > Thanks very much for your very detailed reply. I appreciate you including the additional results and thoroughly answering all of my questions.

---

> > > ### Author Response · Authors · 2023-11-22
> > >
> > > Thank you very much for your review and response!

---

### Official Review · Reviewer_BscE · 2023-10-30

**Soundness:** 3 good
**Presentation:** 3 good
**Contribution:** 2 fair
**Rating:** 5
**Confidence:** 4

**Summary:**

This paper builds on top of Energy Discrepancy, a recently proposed approach for training energy-based models (EBMs), and demonstrates how to leverage Energy Discrepancy to train discrete EBMs. It presents three different types of perturbations, investigates two different types of proposal distributions for efficiently estimating the energy discrepancy loss using importance sampling, and empirically demonstrates the effectiveness of the proposed approach for multiple applications.

**Strengths:**

* The paper studies an important problem: how to efficiently learn discrete EBMs.
* The paper is clearly written, and easy to follow. The method and the experimental results seem sound.
* The paper demonstrates improved performance for a variety of different discrete EBMs.

**Weaknesses:**

All of the proposed perturbation methods and all of the experiments are only dealing with EBMs with binary variables. As a result, presenting the proposed method as an approach to learn discrete EBMs seems to be overclaiming. Moreover, it seems to me the study of this approach when applied to EBMs with discrete variables having more than two states should be within scope of this paper, as this is a very natural and straightforward extension but there might be new issues arising when applied to discrete EBMs with non-binary variables (e.g. the mean-pooling perturbation depends on the parametrizations of the variable states and this can impact the performance of such perturbations).

Related to the above point, I also find the contributions of this paper to be underwhelming. This paper is a straightforward application of Energy Discrepancy, the proposed perturbations are only applicable to binary EBMs, and estimation of Energy Discrepancy is also largely simple adaptations of existing methods. The experimental results also seem to be overclaiming (in table 1, EB-GFN performs best on circles, and in many cases the performance difference seems neglible, elg. on pinwheel; in table 2 the authors claim the proposed methods consistently outperforms baselines which does not seem to be the case; in table 3, the authors claim the performance is comparable but both GWG and DULA seem to be significantly better than the proposed methods on static and dynamic MNIST).

Due to the above weaknesses, I am leaning towards rejecting the paper as is.

**Questions:**

* The paper seems to be taking a lot of the proofs from the original Energy Discrepancy paper more or less verbatim (e.g. appendix A and B). Why is this needed? Why can't these be replaced by a simple pointer to the original paper?
* Page 3 first line, p(x|y) should be p_ebm(x|y)
* The notation p_ebm in equation (6) seems a bit confusing to me
* How is the mean pooling calculated exactly? Take the mean in float and then round to either 0 or 1? It seems to be this would cause issues when extended to non-binary variables as the mean in float for non-binary variables is not particularly meaningful/informative in many cases.
* Mean pooling seems to be applicable only to image data? Presenting it as one of 3 perturbations seems a bit odd as it is more specialized and not very generally applicable.

---

> ### Author Response · Authors · 2023-11-20
> **Response to Reviewer BscE (Part 1)**
>
> Thank you very much for your review. We would like to address your mentioned weaknesses and questions as follows:
>
> > All of the proposed perturbation methods and all of the experiments are only dealing with EBMs with binary variables. As a result, presenting the proposed method as an approach to learn discrete EBMs seems to be overclaiming.
> >
>
> **ANSWER:** Thank you for bringing this to our attention. We will make the necessary revisions in the paper and explicitly state our focus on binary EBMs. We would like to emphasise that our proposed methodology can, in principle, be extended to other types of discrete data sets such as spaces of the form $\{0, … K-1\}^d$ like coloured images or text. In this case, the Bernoulli perturbation may be replaced by a different Markov transition density that moves between states in the space $\{0, … K-1\}^d$. Similarly, the mean-pooling transform can be extended to this setting where one picks certain dimensions to be averaged, and the neighbourhood-based perturbation works as presented if an appropriate neighbourhood structure is defined on the discrete space of interest. We plan to explore these extensions in future work and conduct additional experiments to validate their effectiveness, particularly in the case of text data.
>
> > This paper is a straightforward application of Energy Discrepancy, the proposed perturbations are only applicable to binary EBMs, and estimation of Energy Discrepancy is also largely simple adaptations of existing methods. The experimental results also seem to be overclaiming
> >
>
> **ANSWER:** Thank you for your feedback. We will thoroughly revise the paper to ensure that each claim we make is accurately reflected by our numerical experiments. We further want to clarify our contributions and motivation:
>
> Currently, energy discrepancy has two limitations: Firstly, it only relies on Gaussian perturbations, limiting the approach to continuous settings. Secondly, the variance of the contrastive potential cannot be controlled, forcing the practitioner to use a relatively small noise scale which provably limits the expressiveness of energy discrepancy. This motivated us to look for ways to define perturbations in the discrete context and stabilise the estimation of the loss function. Specifically, we are making the following contributions:
>
> 1. **New Objective for Training Energy-Based Models in Discrete Spaces**
>
> In our work, we successfully extend energy discrepancy [1] to train energy-based models in discrete spaces. Our suggested perturbation introduce new ways to define a perturbation, specifically via a deterministic map or via a neighbourhood structure, and allow the perturbation of multiple dimensions at once, demonstrating that perturbations can be chosen much more flexibly than the relatively limited Gaussian perturbations in the original energy discrepancy.
>
> 2. **New method to estimate the energy discrepancy loss**
>
> The Bernoulli perturbation introduces more noise than the one-bit perturbations used in ratio matching [2, 3] or Gibbs with Gradients [4]. The introduction of importance sampling and gradient guidance helps the reduction of the variance in the contrastive potential in this case and introduces a novel interpretation of contrastive samples that was missing in the original work on energy discrepancies. This approach presents a more general framework compared to the original energy discrepancy paper, which relied on the symmetric properties of Gaussian perturbation for $U_q$ estimation. We anticipate that this new perspective will pave the way for future research in estimating energy discrepancy loss.
>
> While our approach may seem simple, it shows effective in training discrete EBMs, a challenging task that other methods such as ratio matching [2], gradient-guided ratio matching [3], and variational approaches [5] struggle with, even in toy scenarios like grey-code density estimation. Furthermore, our methods accelerate conventional sequential MCMC methods [4, 6] with competitive results.
>
> [1] Schröder, Tobias, et al. "Energy Discrepancies: A Score-Independent Loss for Energy-Based Models." *arXiv preprint arXiv:2307.06431* (2023).
>
> [2] Hyvärinen, Aapo. "Some extensions of score matching." *Computational statistics & data analysis* 51.5 (2007): 2499-2512.
>
> [3] Liu, Meng, Haoran Liu, and Shuiwang Ji. "Gradient-Guided Importance Sampling for Learning Binary Energy-Based Models." ICLR 2023.
>
> [4] Grathwohl, Will, et al. "Oops i took a gradient: Scalable sampling for discrete distributions." *International Conference on Machine Learning*. PMLR, 2021.
>
> [5] Dai, Hanjun, et al. "Learning discrete energy-based models via auxiliary-variable local exploration." *Advances in Neural Information Processing Systems* 33 (2020): 10443-10455.
>
> [6] Zhang, Ruqi, Xingchao Liu, and Qiang Liu. "A Langevin-like sampler for discrete distributions." *International Conference on Machine Learning*. PMLR, 2022.

---

> ### Author Response · Authors · 2023-11-20
> **Response to Reviewer BscE (Part 2)**
>
> > The paper seems to be taking a lot of the proofs from the original Energy Discrepancy paper more or less verbatim (e.g. appendix A and B). Why is this needed? Why can't these be replaced by a simple pointer to the original paper?
> >
>
> **ANSWER:** Thank you for your feedback. We decided to adapt the proofs to the discrete case, specifically, to make them more accessible for a reader with a discrete application in mind. Theorem 2, in particular, was only proven in the Gaussian case in the original Energy Discrepancy paper. In the upcoming revision, we will omit some of the proofs as per your suggestion.
>
> > Page 3 first line, p(x|y) should be p_ebm(x|y). The notation p_ebm in equation (6) seems a bit confusing to me
> >
>
> **ANSWER:** Thank you for pointing out the typo in the first line on page 3. Apologies for the confusion in equation (6). The symbol $p_{ebm}$ in equation (6) represents the energy-based model $p_{ebm}(x) \propto \exp(-U(x))$, and $p_{ebm}(x|y)$ in the first line of page 3 denotes the recovery likelihood $p_{ebm}(x|y) \propto \exp(-U(x))q(y|x)$. We denote both distributions with the same symbol as they assume the same distribution over $x$. We will revise the paper to make it more clear.
>
> > How is the mean pooling calculated exactly? Take the mean in float and then round to either 0 or 1? It seems to be this would cause issues when extended to non-binary variables as the mean in float for non-binary variables is not particularly meaningful/informative in many cases.
> >
>
> **ANSWER:** The mean-pooling perturbation does not involve a rounding operator as the values of the forward perturbation are allowed to take values in a space that is different from the data domain. For the mean pooling perturbation, specifically, the negative samples are generated by randomly permuting the ordering of bits within a given sliding window.
>
> To elaborate, for a deterministic mapping $g: \mathcal{X} \rightarrow \mathcal{Y}$ (e.g., mean pooling), the contrastive potential $U_q(y)$ can be expressed in terms of the preimage of $g$ as
>
> $$
> U_q(y)=-\log \sum_{\{\tilde{x}: g(\tilde{x})=y\}} \exp(-U(\tilde{x}))= -\log \mathbb{E}_{\tilde{x} \sim \mathcal{U}(\{g^{-1}(y)\})}[\exp(-U(\tilde{x}))] + c
> $$
>
> where $g^{-1}(y) = \{\tilde{x}: \tilde{x} \in g^{-1}(y)\}\subseteq \mathcal X$ denotes the preimage of  $y$ under $g$. For the mean-pooling transform, $\{\tilde{x}: \tilde{x} \in g^{-1}(y)\}$ represents a set of points obtained from a permutation of  entries of the data point $x$ within the sliding window, because all these points average to the same value within the sliding window.
>
> The methodology can be extended to non-binary spaces. One first chooses a suitable mapping with a tractable preimage that averages over features in the data vector. Discrete Energy Discrepancy is then computed by uniformly sampling from this preimage.
>
> > Mean pooling seems to be applicable only to image data? Presenting it as one of 3 perturbations seems a bit odd as it is more specialized and not very generally applicable.
> >
>
> **ANSWER:** Apologies if this seemed misleading. The mean pooling perturbation was indeed motivated by the idea to define perturbations for images but it is not limited to that setting; for other types of discrete data vectors, one can still define a sliding window in which the entries are pooled. In Figure 4, we demonstrate the effectiveness of mean pooling in grey-code density estimation. Additionally, we investigate the impact of the window size in figure 6. Surprisingly, we find that the mean pooling transform works better in the density estimation experiments than on images.
>
> We illustrated the mean-pooling transformation using image data for ease of understanding. Generally speaking, the mean pooling transform emphasises that deterministic maps are admissible perturbations in energy discrepancy, a possibility that we felt was not apparent from the original energy discrepancy paper.

---

> ### Comment · Reviewer_BscE · 2023-11-22
> **Thanks for the response**
>
> I thank the authors for the response. I understand that in principle the approach can be extended to the non-binary case, but in practice there can be many potential issues that would come up when we actually try to apply the method to the non-binary case. So without actual empirical experiments and with just some simple discussions I still consider the paper as overclaiming. I suggest the authors either rephrase the paper to be just for binary data or add actual experiments to demonstrate the effectiveness of non-binary data. Additionally I still see the contribution as underwhelming (both in terms of methodology and in terms of empirical results). I would keep my score as is.

---

### Official Review · Reviewer_g3dj · 2023-10-30

**Soundness:** 2 fair
**Presentation:** 3 good
**Contribution:** 2 fair
**Rating:** 3
**Confidence:** 5

**Summary:**

This paper discusses the concept of energy discrepancy application in training energy-based models on discrete data by evaluating the energy function at data points and perturbed counterparts. It is claimed that the paper introduces a novel approach with theoretical guarantees to achieve this goal. Some experiments are provided to support their claim.

**Strengths:**

- This is a very interesting problem to investigate.
- The applicability of this method to perturbed processes makes it versatile for various data domains.
- It is easy to read the paper and the presentation is clear.

**Weaknesses:**

- The novelty of this paper is moderate at best. It seems like this paper is a combination of a few papers for instance:

     * Gradient_Guaided importance sampling for learning binary energy-based models by Meng Liu, Haoran Liu, Shuiwang Ji
     *  Energy Discrepancies: A Score-Independent Loss for Energy-Based Models by Tobias Schröder 2023

- Seems like the theorems are not novel and are borrowed from other papers; they should have been as a proposition with reference to the original paper. Even the proofs are very similar and all could have been omitted.

- This paper seems to be an extension of a poster However, not much is added to the paper compared to the poster. More detailed experimental results are much needed. A lack of comparison between this method and other state-of-the-art on diverse datasets.

**Questions:**

A few questions regarding the algorithm:

1- Seems like sampling a large number of negative samples for each data point in every iteration can be computationally intensive, especially for high-dimensional discrete data which can lead to slowing down the training process. Also, there are many hyperparameters involved in this algorithm such as the number of negative samples, and stabilization parameters. Could the author please elaborate on these concerns? How does the tuning work and how sensitive it is to suboptimal choices?

2-The algorithm's exploration of the energy landscape might be constrained, particularly when the perturbation distribution (q) does not encompass diverse regions of the data space. This limited exploration could lead to models capturing only specific facets of the data distribution, potentially resulting in biased or incomplete representations. Could the authors please elaborate on that? In addition, the choice of perturbation distribution (q) seems to heavily rely on the characteristics of the training data. How can one design an effective perturbation strategy that is adaptive to different types of discrete?

3- Could the authors please discuss the scalability of the proposed methods with increasing dataset sizes? Moreover, can you provide insights into the interpretability of the learned models?

---

> ### Author Response · Authors · 2023-11-20
> **Response to Reviewer g3dj (Part 1)**
>
> Thank you for your thorough review. We are going to reply to your concerns in two parts.
>
> > The novelty of this paper is moderate at best. It seems like this paper is a combination of a few papers for instance
> >
> > - Gradient_Guaided importance sampling for learning binary energy-based models by Meng Liu, Haoran Liu, Shuiwang Ji
> > - Energy Discrepancies: A Score-Independent Loss for Energy-Based Models by Tobias Schröder 2023
>
> **ANSWER:** Thank you for your comment. We want to clarify our contribution and motivation:
>
> Currently, energy discrepancy has two limitations: Firstly, it only relies on Gaussian perturbations, limiting the approach to continuous settings. Secondly, the variance of the contrastive potential cannot be controlled, forcing the practitioner to use a relatively small noise scale which provably limits the expressiveness of energy discrepancy. This motivated us to look for ways to define perturbations in the discrete context and stabilise the estimation of the loss function. We propose techniques to address these limitations of the original energy discrepancy and to target the challenging problem of training EBMs on discrete domains, a research area with only a relatively small number of existing methodologies.
>
> We believe that new methods for discrete energy-based modelling are needed. Maximum-Likelihood based techniques always rely on a sequentially operating samplers that are evaluated after a short number of steps, resulting in biased energy estimates. Gradient-Guided Ratio Matching, on the other hand, is limited to proposals that flip a single dimension. Furthermore, it has not been demonstrated that Gradient Guided Ratio Matching works on image data sets.
>
> Discrete energy discrepancy offers a simple, effective, and cheap to evaluate alternative to these training methodologies. Specifically, we are making the following contributions:
>
> 1. **New Objective for Training Energy-Based Models in Discrete Spaces**
>
>     In our work, we successfully extend energy discrepancy to train energy-based models in discrete spaces. Our suggested perturbation introduce new ways to define a perturbation, specifically via a deterministic map or via a neighbourhood structure, and allow the perturbation of multiple dimensions at once, demonstrating that perturbations can be chosen much more flexibly than the relatively limited Gaussian perturbations in the original energy discrepancy.
>
> 2. **Variance Reduction in Energy Discrepancy Loss Estimation**
>
>     The Bernoulli perturbation introduces more noise than the one-bit perturbations used in ratio matching or GwG. For this reason, a low variance estimator of the contrastive potential $U_q$ is critical to the success of energy discrepancy. The introduction of importance sampling and gradient guidance helps the reduction of the variance in the contrastive potential in this case and introduces a novel interpretation of contrastive samples that was missing in the original work on energy discrepancies. This approach presents a more general framework compared to the original energy discrepancy paper, which relied on the symmetric properties of Gaussian perturbation for $U_q$ estimation. We anticipate that this new perspective will pave the way for future research in estimating energy discrepancy loss.
>
> > Seems like the theorems are not novel and are borrowed from other papers; they should have been as a proposition with reference to the original paper. Even the proofs are very similar and all could have been omitted.
> >
>
> **ANSWER:** Thanks for your suggestions. The proof of theorem 2 had to be adapted to establish that the theorem holds for the newly proposed perturbations and for the gradient guidance scheme. We will remove some other proofs in our revision to make new contributions easier to identify.
>
> > [...] More detailed experimental results are much needed. A lack of comparison between this method and other state-of-the-art on diverse datasets.
> >
>
> **ANSWER:** Training energy-based models in discrete spaces poses a significant challenge. The testing of discrete EBM training algorithms in previous literature generally involves four standard benchmarks: Ising models, grey-code density estimation, ego-graph generation, and discrete image modelling, see e.g. [1, 2]. Our methods achieve competitive results with a simpler and faster training routine and better theoretical guarantees in these benchmarks. Despite performance gaps compared to MCMC-based methods on image modelling tasks, one should note that our approaches do not depend on MCMC and can outperform CD algorithms up to five MCMC steps by a considerable margin.
>
> [1] Grathwohl, Will, et al. "Oops i took a gradient: Scalable sampling for discrete distributions." International Conference on Machine Learning. PMLR, 2021.
>
> [2] Zhang, Dinghuai et al. "Generative Flow Networks for Discrete Probabilistic Modeling" International Conference on Machine Learning. PMLR, 2022

---

> ### Author Response · Authors · 2023-11-20
> **Response to Reviewer g3dj (Part 2)**
>
> > Seems like sampling a large number of negative samples for each data point in every iteration can be computationally intensive, especially for high-dimensional discrete data which can lead to slowing down the training process. Also, there are many hyperparameters involved in this algorithm such as the number of negative samples, and stabilization parameters. Could the author please elaborate on these concerns? How does the tuning work and how sensitive it is to suboptimal choices?
> >
>
> **ANSWER:** Thanks, for raising these concerns, we are happy to elaborate. In this work, we use three hyperparameters: The number of negative samples, one stabilisation parameter, and the choice of perturbation. We would like to remark that the number of hyperparameters is not larger than in contrastive divergence methods which rely on tuning the MCMC subroutine.
>
> Our approach is scalable in the number of negative samples. In all experiments, we selected the number of negative samples as $M=32$ irrespective of the dimension of the problem to maximise $M$ within the constraints of our GPU capacity. We demonstrate in Figures 7, 8, and 9 in the appendix that our results remain largely unchanged when choosing the number of negative samples as smaller or larger. To further investigate, we train energy-based models using different values of $M$ on the static MNIST dataset with ED-Grid. The results show that our results remain of comparable quality as the number of negative samples is decreased. Furthermore, our approach is easier to parallelise than the sequentially computed MCMC of contrastive divergence.
>
> |  | M=4 | M=8 | M=16 | M=32 |
> | --- | --- | --- | --- | --- |
> | NLL | 90.13 | 90.37 | 89.14 | 90.61 |
>
> Concerning the stabilisation parameter $w$ and the noise scalar of Bernoulli perturbation $\epsilon$, we conducted a comprehensive ablation study presented in figures 3, 5, 6, 7, 8, and 9. Our training method is robust in both parameters and not sensitive to suboptimal choices. Throughout all experiments, no specific techniques were employed for hyperparameter tuning and search. Further details regarding the hyperparameter setup are provided in Appendix D.
>
> > The algorithm's exploration of the energy landscape might be constrained, particularly when the perturbation distribution (q) does not encompass diverse regions of the data space. This limited exploration could lead to models capturing only specific facets of the data distribution, potentially resulting in biased or incomplete representations. Could the authors please elaborate on that? In addition, the choice of perturbation distribution (q) seems to heavily rely on the characteristics of the training data. How can one design an effective perturbation strategy that is adaptive to different types of discrete?
> >
>
> **ANSWER:** Thanks for your comment. The selection of perturbation is indeed pivotal in training discrete EBMs with energy discrepancy. Theoretically the introduction of more noise by $q(y\vert x)$ enhances the expressiveness of energy discrepancy but concurrently destabilises training. Hence, there exists a trade-off between exploration and training stability.
>
> The perturbation choices introduced in this work adhere to the conditions of Theorem 1 and thus lead to unbiased estimates of the data density, at least in theory. Furthermore, all perturbations are guaranteed to explore the state space close to data points and hence regions with high likelihoods are represented faithfully. This is supported by our experiments which show that simply perturbing the data and learning from the contrast without relying on heavy MCMC sampling can yield successful results, at least on those four standard benchmarks. This stands out as the primary contribution and insight of our work. Moreover, the Bernoulli perturbation has the theoretical capacity to cover all regions of the data space since it allows the flipping of all bits at random to construct negative samples.
>
> Despite this success, we don't expect that one single choice of perturbation can work for all data sets, which is likely a trade-off that one needs to make to give up on the expensive MCMC subroutine used in most training methods for energy-based models. Generally speaking, the perturbation should be informed by the type of data and should destroy information effectively (e.g. a Markov process on non-binary discrete data), while the proposal should be adaptive to the energy and put emphasis on states that require adjustments of the energy to achieve low-variance estimates of the contrastive potential. One way to achieve the adaptive component of this strategy is using the gradient guidance sampling proposed in our paper. For example, it can be seen that ED-$\nabla$Bern exhibits significant performance gains compared to ED-Bern in Table 3 and outperforms CD-1 by a considerable margin. Exploring other means for low variance estimates of the energy discrepancy loss remain a promising avenue for future research.

---

> ### Author Response · Authors · 2023-11-20
> **Response to Reviewer g3dj (Part 3)**
>
> > Could the authors please discuss the scalability of the proposed methods with increasing dataset sizes? Moreover, can you provide insights into the interpretability of the learned models?
> >
>
> **ANSWER:** Regarding the number of data points, the loss is computed and optimised in mini-batches, so there are no unusual constraints in its scalability. Regarding the dimension of the data set, the major constraint in high dimensions are singular data supports which do not admit a representation by a positive density or energy function. This problem is universal to energy-based models, see e.g. [3], making training of energy-based models particularly challenging compared to other types of generative models. This also poses some constraints to apply energy-based models to data sets of arbitrary complexity, for this reason we focussed on standard benchmarks in our work.
>
> Interpreting energy-based models is challenging because the best metric to evaluate the learned model are it's log-likelihood or it's sample quality, but the two metrics often yield contradicting results. Based on the theoretical guarantees from Theorem 1 and our results on synthetic density estimation, we expect that energy discrepancies match the data density more accurately than standard methods like contrastive divergence.
>
> [3] Arbel et al. "Generalized Energy-Based Models" ICLR 2021

---

> ### Comment · Reviewer_g3dj · 2023-11-22
>
> I'd like to thank the authors for their response. I have read the responses and other reviews. I appreciate the new details provided by the authors, however, I will keep my score.

---

### Official Review · Reviewer_tPh7 · 2023-11-01

**Soundness:** 3 good
**Presentation:** 3 good
**Contribution:** 2 fair
**Rating:** 6
**Confidence:** 3

**Summary:**

The paper generalizes the energy discrepancy training algorithm from continuous space to discrete space for training energy based models. In particular, the paper introduces three types of perturbations to construct the energy discrepancy in discrete space. Also, the paper developed gradient-based proposals in discrete space to efficiently estimate the energy discrepancy. The paper conducts experiments on Ising model, synthetic discrete patterns, graph generation, and discrete images. The results shows the proposed method is a competitive algorithm compared to existing energy based models training algorithms in discrete space.

**Strengths:**

* The paper, in the first time, introduces energy discrepancy objective in training discrete energy based models. The idea is simple but smart, and looks promising especially for training deep energy models where a disturbed data is supposed to be a good negative sample.
* The paper conducts intensive experiments with different perturbations and gradient-based proposal methods to justify the effectiveness of the proposed method. Also mentioned some basic philosophy to choose perturbation and proposal, for example, in section 6.4, the paper conjectures the combination of gradient-based proposal with grid neighborhood transformation tends to get trapped in local modes as it only
flips one bit for each negative sample.
* The paper is well written with appropriate explanation and examples.

**Weaknesses:**

* The energy discrepancy was used to train continuous energy based models. The paper does not provide enough evidence about the limitation of continuous perturbation with continuous relaxation to train discrete energy based models. This makes the paper less motivated. An explanation about why continuous perturbation does not work or some empirical comparisons would be helpful.
* Although the paper provides comparisons between different perturbation and proposals, a principled criteria to choose them is missing. This is no doubt a hard question and a bit more discussion would be helpful.
* In experiments, the proposed training algorithm is not significantly better than existing methods. I guess the main reasons are the tasks are simple and the existing algorithms already get very good results. As mentioned in the last paragraph of the paper, the evaluations on more structured data like molecules or text would be helpful.

Minor:\
Clarification in page 2, theorem 1. $\text{Var}(x|y)$ is a bit confusing. $\text{Var} (z), z \sim p(\cdot|y)$ would be easier to understand for me.\
Typo in page 3, first sentence: $p(x|y) \Rightarrow p_\text{ebm}(x|y)$

**Questions:**

* Will using an energy based model with continuous relaxation with continuous energy discrepancy training work? How does it perform compared to current discrete framework?
* Can we train a variational distribution for informed proposal? How would that compare to a gradient-based proposal?

---

> ### Author Response · Authors · 2023-11-19
> **Comments on Mentioned Weaknesses**
>
> Thank you very much for your review and your comments! Here are our responses to the mentioned weaknesses and your questions:
>
> First of all, thank you for making us aware of typos and some unclear points in the notation. We will correct them in our revision.
>
> > The energy discrepancy was used to train continuous energy based models. The paper does not provide enough evidence about the limitation of continuous perturbation with continuous relaxation to train discrete energy based models. This makes the paper less motivated. An explanation about why continuous perturbation does not work or some empirical comparisons would be helpful.
> >
>
> **ANSWER:** Thank you for raising this concern. We have conducted a number of experiments on binary data sets using the Gaussian perturbation, which was previously used when applying energy discrepancy to continuous data. Our experiments show that Gaussian energy discrepancies are incapable of training energy-based models on discrete data. We will include plots of the failure case in our revision. This motivates the development of new perturbations that are tailored to estimation tasks on discrete data sets.
>
> > Although the paper provides comparisons between different perturbation and proposals, a principled criteria to choose them is missing. This is no doubt a hard question and a bit more discussion would be helpful.
> >
>
> **ANSWER:** Thanks for your intriguing question. There is a trade-off between the expressiveness of energy discrepancy and training stability. The original energy discrepancy paper suggests that the more noise is introduced by $q(y\vert x)$, the more expressive the energy discrepancy loss becomes. However, introducing too much noise destabilises training at the same time.
>
> In this work, we target both aspects of this trade-off. Firstly, we introduce three different perturbations: The grid neighbourhood perturbs only one bit at once which allows for stable training and good performance on image data sets, but leads to a relatively small exploration of the state space. The Bernoulli perturbation, in contrast, enables the flipping of multiple bits at once which is believed to result in a more expressive energy discrepancy and faster learning.
>
> In turn, our proposition of gradient guidance reduces the variance of the estimated energy discrepancy loss without giving up its theoretical guarantees. In addition, the gradient-guided negative samples explore states with low energies more often which means that the contrastive samples are more informative about how the energy landscape needs to be adjusted to match the given data distribution. The resulting update then resembles the update of Contrastive divergence with Gibbs with Gradients [1] or Discrete Langevin [2], but with better theoretical guarantees.
>
> To summarise, the perturbation should be chosen to introduce noise, while the proposal should be chosen to reduce the variance of the energy discrepancy loss function. We will add a short discussion about the effect of various choices for the perturbation and proposal.
>
> > In experiments, the proposed training algorithm is not significantly better than existing methods. I guess the main reasons are the tasks are simple and the existing algorithms already get very good results. As mentioned in the last paragraph of the paper, the evaluations on more structured data like molecules or text would be helpful.
> >
>
> **ANSWER:** Thanks for your suggestions. Training energy-based models in discrete spaces poses challenging and current state-of-the-art methods, including GwG [1] and DULA [2], have also been tested on relatively simple datasets like static MNIST and Omniglot. The challenges of learning EBMs over discrete data sets persist even in synthetic tasks such as grey-code density estimation, where previous approaches like ratio matching encounter difficulties.
>
> The experiments conducted by us are standard in the literature on discrete energy-based model training (see GwG [1], DULA [2] GNetFlow EBMs [3], and RMwGGIS [4]). Our methods exhibit particularly good performance in training Ising models, grey-code density estimation, and ego graph generation, where they produce competitive results to other training methods at a smaller computational cost. Despite performance gaps compared to MCMC-based methods in discrete image modelling, our approaches offer better theoretical properties, faster training, and can outperform CD-1 by a considerable margin.
>
> Nevertheless, we believe that assessing our methods on more structured data such as molecules or text would be beneficial. This would not only make our methods more meaningful but also introduce additional benchmarks for discrete energy-based models. We hope that this can be achieved through better perturbation and proposal choices, and consider this as a potential avenue for future research.

---

> ### Author Response · Authors · 2023-11-19
> **Responses to Questions, References**
>
> These are the remaining answers to your questions:
>
> > Will using an energy based model with continuous relaxation with continuous energy discrepancy training work? How does it perform compared to current discrete framework?
> >
>
> **ANSWER:** We were unable to replicate any of our presented experiments on discrete data sets with the originally proposed Gaussian energy discrepancy. We have tried multiple choices for the hyper-parameters. We will add a figure to our revision which shows our most successful attempt (Figure 5), which just barely reflects some structure of the data set the EBM was trained on.
>
> > Can we train a variational distribution for informed proposal? How would that compare to a gradient-based proposal?
> >
>
> **ANSWER:** This is certainly an exciting idea to reduce the variance of the contrastive potential further. The biggest caveat with variational approaches are potential instabilities from the resulting minimax game. Furthermore, a variational proposal adds additional complexity to the methodology like choosing suitable architectures and optimisation schemes, as well as additional computational costs. We are interested in exploring variational approaches in a separate work.
>
> **References**
> [1] Grathwohl, Will, et al. "Oops i took a gradient: Scalable sampling for discrete distributions." *International Conference on Machine Learning*. PMLR, 2021.
>
> [2] Zhang, Ruqi, Xingchao Liu, and Qiang Liu. "A Langevin-like sampler for discrete distributions." *International Conference on Machine Learning*. PMLR, 2022.
>
> [3] Zhang, Dinghuai et al. "Generative Flow Networks for Discrete Probabilistic Modeling" *International Conference on Machine Learning*. PMLR, 2022
>
> [4] Liu, Meng, et al. "Gradient-Guided Importance Sampling for Learning Binary Energy-Based Models." ICLR 2023

---

> > ### Comment · Reviewer_tPh7 · 2023-11-20
> > **Response from Reviewer tPh7**
> >
> > After reading the rebuttal, I appreciate the authors' great effort in responding and revising. I believe the revised version is more clear and I will keep my score.

---

> > > ### Author Response · Authors · 2023-11-20
> > >
> > > Thank you for your review and for your appreciation of our efforts. We will upload the revised version of the paper soon.

---

### Official Review · Reviewer_wato · 2023-11-06

**Soundness:** 2 fair
**Presentation:** 3 good
**Contribution:** 2 fair
**Rating:** 5
**Confidence:** 4

**Summary:**

This research suggests a modification to the training process of Energy-Based Models (EBMs) by using the energy discrepancy loss. The main idea is to replace EBMs' conventional negative samples with perturbed versions of the observed samples, generared by a conditional distribution $q(y∣x)$. The energy discrepancy, which has previously been used in the modeling of continuous distributions, is here extended to address discrete distributions. The paper offers several designs for the perturbation mechanism $q$, including the application of Bernoulli noise, deterministic transformations, and neighborhood structures.

**Strengths:**

1. The paper presents its findings in a clear and structured manner, offering clear explanations of the energy discrepancy loss and provide useful insights on its connection to prior loss functions.

2. The paper focuses on the modeling of discrete distributions using Energy-Based Models (EBMs), an area that has not been extensively explored in previous research.

**Weaknesses:**

While the authors assert that their proposed model can provide flexibility and theoretical guarantees in the construction of negative samples without the need of MCMC sampling. After reading the paper, I still hold some doubts on whether this methods is powerful and flexible enough in modeling complex distributions. My concerns and questions are as follows:

1. Influence of the gap between KL Divergence and KL-Contraction Divergence on estimation

Equation (6) in the paper suggests that minimizing energy discrepancy loss is equivalent to minimizing KL-contraction divergence. Nevertheless, the KL-contraction divergence serves merely as a lower bound of the KL divergence, where the divergence between two transformed distributions, indicated by $q(y∣x)$, defines this bound. If we consider an extreme scenario where $q(y∣x)$ becomes (or is very close to) a deterministic identity function, then the transformed distributions may closely resemble the original distributions. Consequently, the KL-contraction could approach zero, irrespective of a potentially significant disparity between the actual data distribution and the fitted model. This raises my concern that the proposed loss term might offer a bound too lax to guide the optimization effectively toward the equivalence of $p_1$ and $p_2$.

2. Whether there can be a dilemma in selecting $q(y|x)$?

Building on my initial point, securing a tighter lower bound might necessitate a $q(y∣x)$ that significantly alters x by losing enough information. At another extreme, selecting $q(y∣x)$ that maps all samples to a normal distribution, irrespective of x (akin to infusing substantial noise as in diffusion processes), would mean the KL-contraction mirrors the original KL divergence closely. However, an excessive deviation of y from x (transforming it to what resembles random noise) could render negative samples trivially distinguishable from positive ones, potentially destabilizing the training. This seems to create a dilemma. Although in the experiments shown in the paper, the proposed simple perturbations seem to work. Given that the distributions shown in the paper are not very complex, I'm concerned about this problem in modeling more complex distributions.


 3.  The Feasibility of Abandoning MCMC Sampling for Complex Distributions:

The discussion in Section 4.1 of the paper introduces an importance sampling strategy that utilizes gradient information (as shown in equation 12) to stabilize training. This approach seems to parallel a single-step gradient-based MCMC sample that 'denoises' the perturbed data (so that the negative samples become closer to true samples). I question whether such a simplified rule is viable for more convoluted distributions, or if multiple updates might be necessary to yield valid samples, essentially reverting to the conventional MCMC technique.

4. Sufficiency of Experimental Results to Substantiate the Algorithm's Effectiveness:

Reflecting upon the earlier points, the experiments demonstrated in this paper seem to be a bit easy to me. While models like the Ising Model and 2D examples can serve as sanity check examples, they might not pose a significant challenge with the current state of research. Similarly, the Ego-small dataset might lack complexity in its graph structure . Moreover, reference [1] seems to set a more robust benchmark. The MNIST dataset, used for image distribution analysis, is also relatively simplistic, and it appears the proposed model falls short of matching the performance of GWG. These factors collectively cast doubt on the proposed algorithm's performance and scalability when dealing with more sophisticated distributions.

[1] Score-based Generative Modeling of Graphs via the System of Stochastic Differential Equations.

5. More comprehensive review for EBM works:

While the paper primarily concentrates on modeling discrete distributions, it would be beneficial to provide a more comprehensive review of the field of Energy-Based Model (EBM) training, with particular emphasis on those studies that explore sampling methods for negative samples. For example, [1] pioneering work using cnn as energy function, [2] [3] [6] are works that use different armortized methods to reduce MCMC steps, [4] uses a replay buffer and [5] moves to the latent space to facilitate sampling.

[1] "A theory of generative convnet." International Conference on Machine Learning. PMLR, 2016.

[2]  "Cooperative training of descriptor and generator networks." IEEE transactions on pattern analysis and machine intelligence 2018

[3]  "No MCMC for me: Amortized sampling for fast and stable training of energy-based models." ICLR 2021

[4] "Improved contrastive divergence training of energy-based models." ICML 2021

[5] . "VAEBM: A Symbiosis between Variational Autoencoders and Energy-based Models" ICLR 2021

[6] "A tale of two flows: Cooperative learning of langevin flow and normalizing flow toward energy-based model." ICLR 2022

**Questions:**

Please check the weakness part.

---

> ### Author Response · Authors · 2023-11-19
> **Response to Reviewer wato (Part 1)**
>
> Thank you very much for your constructive comments. Here is our response to your questions:
>
> > 1. Influence of the gap between KL Divergence and KL-Contraction Divergence on estimation.
> >
> > [...] the KL-contraction could approach zero, irrespective of a potentially significant disparity between the actual data distribution and the fitted model. This raises my concern that the proposed loss term might offer a bound too lax to guide the optimization effectively toward the equivalence of $p_1$ and $p_2$.
>
> **ANSWER:** It is true that the KL contraction serves as a lower bound of the KL divergence, and if $q(y \vert x)$ becomes the identity, the KL contraction shrinks to zero. However, since the perturbation is not adapted throughout training but chosen in advance, this can be prevented by choosing a perturbation that discriminates between two distributions effectively.
>
> In all settings introduced in our work, the perturbation is significantly different from the identity function and introduces the contrast between positive and negative samples needed to learn the appropriate energy function. This is underlined by Theorem 1, which states that energy discrepancy has a unique minimiser at $\exp(-U^\ast) \propto p_{\mathrm{data}}$ under mild conditions that are satisfied for the Bernoulli and neighbourhood-based perturbation. Furthermore, our experimental results support the claim that discrete energy discrepancy can learn complex data distributions like binary image data sets.
>
> Furthermore, the original energy discrepancy paper [7] shows that contrastive divergence can be interpreted as an energy discrepancy with adaptive perturbation $q(y\vert x)$, which may shrink to zero as contrastive divergence converges. In this sense, discrete energy discrepancy is not more lax than standard training methods for energy-based models.
>
> > 2. Whether there can be a dilemma in selecting $q(y\vert x)$?
> >
> > [...] securing a tighter lower bound might necessitate a $q(y\vert x)$ that significantly alters x by losing enough information. [...] However, an excessive deviation of y from x (transforming it to what resembles random noise) could render negative samples trivially distinguishable from positive ones, potentially destabilizing the training.
>
> **ANSWER:** There is indeed a trade-off between the expressiveness of energy discrepancy and the training stability. The more noise is introduced by $q(y\vert x)$, the more expressive the loss becomes while destabilising training at the same time. This trade-off guides the design choices made by us in this paper. The grid-neighbourhood perturbation is a relatively small perturbation which allows stable training and produces good results. The Bernoulli perturbation is more ambitious, as it flips several dimensions at once and leads to a more discriminative albeit less stable loss function. We employ gradient guidance specifically to reduce the variance of the contrastive potential and estimated loss function, thus allowing for more complex perturbations. Our results show that gradient guidance indeed improves the performance of Bernoulli-based energy discrepancy.
>
> Given the difficulty of training energy-based models on discrete spaces, we believe that the choice of $q$ is an acceptable trade-off to be made by the practitioner.
>
> > 3. The Feasibility of Abandoning MCMC Sampling for Complex Distributions.
> >
> > The discussion in Section 4.1 of the paper introduces an importance sampling strategy that utilizes gradient information [...]. This approach seems to parallel a single-step gradient-based MCMC sample that 'denoises' the perturbed data [...]. I question whether such a simplified rule is viable for more convoluted distributions, or if multiple updates might be necessary to yield valid samples, essentially reverting to the conventional MCMC technique.
>
> **ANSWER:** The gradient guidance is indeed similar to contrastive divergence with one step of MCMC. However, we also weight the contribution of each contrastive sample with an appropriate importance weight. This removes the biases of conventional contrastive divergence, since energy discrepancy, contrary to conventional MCMC techniques, offers theoretical guarantees. Indeed, we observe that conventional MCMC techniques only become competitive when more than seven steps of MCMC are used, a procedure that is more expensive than evaluating energy discrepancy. We report the negative likelihoods of MCMC-based techniques with k steps (CD-k) and energy discrepancy in table six (lower is better):
>
> | Dataset \Method | CD-1 | CD-7 | CD-10 | ED-Bern | ED-$\nabla$Bern |
> | --- | --- | --- | --- | --- | --- |
> | Static MNIST | 182.53 | 98.07 | 88.13 | 96.11 | 90.16 |
>
> We have not conducted experiments on more involved distributions. Approximating the optimal importance distribution with more gradient steps might also improve the performance of energy discrepancy in these cases, which we leave as an avenue for future research.

---

> ### Author Response · Authors · 2023-11-19
> **Response to Reviewer wato (Part 2)**
>
> > 4. Sufficiency of Experimental Results to Substantiate the Algorithm's Effectiveness.
> >
> > Reflecting upon the earlier points, the experiments demonstrated in this paper seem to be a bit easy to me. [...] Moreover, reference [1] seems to set a more robust benchmark. The MNIST dataset, used for image distribution analysis, is also relatively simplistic, and it appears the proposed model falls short of matching the performance of GWG. These factors collectively cast doubt on the proposed algorithm's performance and scalability when dealing with more sophisticated distributions.
> >
>
> **ANSWER:** Training energy-based models in discrete spaces is a challenging problem. Existing leading methods, such as contrastive divergence with Gibbs with Gradients [2] and Langevin for discrete distribution [3] are primarily tested on simple datasets like static MNIST and Omniglot as well. Similarly, the other experiments on Ising models, synthetic grey-code data, and ego-small graphs have been used as test cases for discrete energy-based model training, see e.g. [2] and [4]. Compared to contrastive divergence, our approach works without MCMC which means that energy discrepancy can simplify and accelerate the training of energy-based models on typically benchmarked data sets.
>
> The cited reference [1] uses a diffusion model, which serves different purposes compared to energy-based models as they learn a sampler for the data distribution, while energy-based models model the data generating density, directly.
>
> We agree that our models seem to fall short compared to GwG in our experiments on image data. However, our training method outperforms contrastive divergence with up to seven steps of GwG, while being significantly cheaper to evaluate. We hope that future developments of energy discrepancy can match the performance of GwG at a lower computational cost.
>
> > 5. More comprehensive review for EBM works.
> >
> > While the paper primarily concentrates on modeling discrete distributions, it would be beneficial to provide a more comprehensive review of the field of Energy-Based Model (EBM) training, with particular emphasis on those studies that explore sampling methods for negative samples.
> >
>
> **ANSWER:** Thank you for the diverse list of references, they are a useful resources for exploring sampling methods to generate negative samples! It should be noted that most mentioned methods use negative samples for approximate maximum likelihood estimation, which suggests different methods to produce negative samples. We will include more references including these in the related work section. Thank you for your contribution.
>
> **References:**
> [1] Score-based Generative Modeling of Graphs via the System of Stochastic Differential Equations.
>
> [2] Grathwohl, Will, et al. "Oops i took a gradient: Scalable sampling for discrete distributions." International Conference on Machine Learning. PMLR, 2021.
>
> [3] Zhang, Ruqi, Xingchao Liu, and Qiang Liu. "A Langevin-like sampler for discrete distributions." International Conference on Machine Learning. PMLR, 2022.
>
> [4] Liu, Meng, Haoran Liu, and Shuiwang Ji. "Gradient-Guided Importance Sampling for Learning Binary Energy-Based Models." ICLR 2023.

---

> ### Author Response · Authors · 2023-11-22
>
> Dear Reviewer wato,
>
> Thank you for reviewing our work and providing valuable feedback. As the discussion period is drawing to a close, we would appreciate it if you could confirm whether our rebuttal adequately addressed your concerns. Please let us know if any issues persist.
>
> Best wishes,
> The authors

---

> > ### Comment · Reviewer_wato · 2023-11-23
> >
> > Dear Authors,
> >
> > Thank you for your responses, which have addressed some of my concerns. However, I maintain some reservations regarding the scalability of this work. And I agree with Reviewer BscE's perspective that additional experimental results on non-binary case could be beneficial. Considering the complexities inherent in sampling within a discrete space, I will not strongly reject the acceptance of this paper, but I might want to keep my score for now.

---

### Author Response · Authors · 2023-11-21
**Revision**

Dear Reviewers,

Thanks for your helpful and constructive comments. We have revised the manuscript accordingly. The main changes are marked in blue.

- Reviewer **wato:** We included the references to continuous EBM training methods in the related work
- Reviewer **tPh7:** We fixed the typo and demonstrated the failure of continuous relaxations on discrete EBM tasks in Appendix C.2. We further discussed briefly how to extend the perturbations to non-binary state spaces and the main principles that guide the choice of perturbation and proposal.
- Reviewer **g3dj:** We excluded some proofs and included the experiment with varying numbers of negative samples in Table 8 in Appendix C.4.
- Reviewer **BscE:** We omitted certain proofs, corrected the typo, and refined the experiment section to ensure that we avoid overstating our claims. Additionally, we sketch how to extend the introduced perturbations to non-binary discrete spaces.
- Reviewer **kTVF:** We included the running time comparison in Table 7 in Appendix C.4.

Best,

The authors

---

### Author Response · Authors · 2023-11-21
**Summary (Part 1)**

We thank all reviewers for their constructive comments, which have significantly contributed to the improvement of our work.

First, we would like to summarise the strengths of the paper according to the reviewers:

The reviewers agree that the development of training methods for energy-based models on discrete spaces is an important problem with little existing literature.

> **wato:** The paper focuses on the modeling of discrete distributions using Energy-Based Models (EBMs), an area that has not been extensively explored in previous research. **g3dj:** This is a very interesting problem to investigate. **BscE:** The paper studies an important problem: how to efficiently learn discrete EBMs.
>

Furthermore, the reviewers find the methodology a promising tool for the training of discrete EBMs.

> **tPh7**: The paper, in the first time, introduces energy discrepancy objective in training discrete energy based models. The idea is simple but smart, and looks promising especially for training deep energy models where a disturbed data is supposed to be a good negative sample. **g3dj:** The applicability of this method to perturbed processes makes it versatile for various data domains. **BscE**: The method and the experimental results seem sound. The paper demonstrates improved performance for a variety of different discrete EBMs.
>

The reviewers consider our paper to be well-written and well-supported by experiments.

> **wato**: The paper presents its findings in a clear and structured manner, offering clear explanations of the energy discrepancy loss and provide useful insights on its connection to prior loss functions. **tPh7**: The paper conducts intensive experiments with different perturbations and gradient-based proposal methods to justify the effectiveness of the proposed method. The paper is well written with appropriate explanation and examples. **g3dj:** It is easy to read the paper and the presentation is clear. **BscE**: The paper is clearly written, and easy to follow. **kTVF**: The authors present thorough ablations and experimental details in the appendix.
>

---

> ### Author Response · Authors · 2023-11-21
> **Summary (Part 2)**
>
> We now summarise the main concerns that were raised by the reviewers and explain how we addressed them in our rebuttal:
>
> 1. **The main contributions and novelty of this paper**:
> Reviewers ****g3dj and BscE**** are concerned about the novelty of the contributions made in our paper. We would like to motivate why we see our contributions as useful to the machine learning community and why they are novel.
>
> Discrete energy-based modelling is a challenging problem with a relatively small number of methodologies. Most methods use sampling techniques to approximate the gradient of the log-likelihood. While these approaches have shown some success, it is known that they are often computationally expensive and offer no theoretical guarantees.
>
> Our adaption of energy discrepancy to the discrete domain has shown surprising success on all tested data sets despite the simplicity of our proposed methodology which requires no MCMC. Discrete energy discrepancy even works on simple image data sets like MNIST, a data set where other estimation techniques like ratio matching as well as the continuous energy discrepancy are known to fail. For this reason, we believe that discrete energy discrepancy is a useful addition to the toolbox of discrete EBM training methodologies.
>
> **To be more precise, we would like to highlight our contributions as follows:**
>
> Currently, energy discrepancy [1] has two limitations: Firstly, it only relies on Gaussian perturbations, limiting the approach to continuous settings. Secondly, the variance of the contrastive potential can not be controlled, forcing the practitioner to use a relatively small noise scale which provably limits the expressiveness of energy discrepancy. This motivated us to look for ways to define perturbations in the discrete context and stabilise the estimation of the loss function. Our suggested perturbation choices are largely novel, allow the perturbation of multiple dimensions at once, and demonstrate that perturbations can be chosen much more flexibly than the relatively limited Gaussian perturbations in the original energy discrepancy paper or the single-bit flips in ratio matching [2].
>
> The Bernoulli perturbation introduces more noise than the one-bit perturbations used in ratio matching or GwG [3]. For this reason, a low variance estimator of the contrastive potential $U_q$ is critical to the success of energy discrepancy. The introduction of gradient-guided importance sampling helps the reduction of the variance in the contrastive potential and introduces a novel interpretation of contrastive samples that was missing in the original work on energy discrepancies. This approach represents a more general framework compared to the original energy discrepancy paper, which relied on the symmetric properties of Gaussian perturbation for $U_q$ estimation. While gradient-guided ratio matching [4] also uses importance sampling and gradient guidance, the proposals are limited to flipping a single entry of the binary data vector. It also seems to us that RMwGGIS is currently limited to synthetic data sets and has not been applied to image data sets, successfully. We hope that this new perspective will pave the way for future research in estimating energy discrepancy losses.

---

> ### Author Response · Authors · 2023-11-21
> **Summary (Part 3)**
>
> 2. **The scalability of the methods proposed in this paper**
>
> Reviewer ****wato, tPh7 and g3dj**** remark that the experiments may not be scalable and that more experimental results should be added.
>
> From our experience, training discrete energy-based models is challenging. For this reason, the complexity of benchmark data sets is often not as high as with continuous data sets or different types of generative models. The experimental results presented in this paper are typical benchmarks for discrete energy-based models, see e.g. previous works on Gibbs with Gradients [3], Gradient-Guided Ratio Matching [4], Discrete Langevin Samplers [5], and Generative Flow Networks for EBM training [6]. Our method is competitive to previous work, while offering theoretical guarantees and a cheap and easy to implement parameter update. In particular, our method outperforms CD-1 [7] and ratio matching with and without gradient guidance. For this reason, we believe that our work covers typical experimental settings well.
>
> In addition, our method is scalable since the number of contrastive samples does not grow prohibitively as we increase the dimension of the training data.
>
> We have been made aware that all of our experiments regard binary discrete spaces rather than arbitrary discrete spaces. We will make this limitation to binary discrete types of data clearer in our revision, and characterise ways to extend our work to non-binary discrete spaces.
>
> ### References:
>
> [1] Schröder, Tobias, et al. "Energy Discrepancies: A Score-Independent Loss for Energy-Based Models." *arXiv preprint arXiv:2307.06431* (2023).
>
> [2] Hyvärinen, Aapo. "Some extensions of score matching." *Computational statistics & data analysis* 51.5 (2007): 2499-2512.
>
> [3] Grathwohl, Will, et al. "Oops i took a gradient: Scalable sampling for discrete distributions." *International Conference on Machine Learning*. PMLR, 2021.
>
> [4] Liu, Meng, Haoran Liu, and Shuiwang Ji. "Gradient-Guided Importance Sampling for Learning Binary Energy-Based Models." **ICLR 2023.
>
> [5] Zhang, Ruqi, Xingchao Liu, and Qiang Liu. "A Langevin-like sampler for discrete distributions." *International Conference on Machine Learning*. PMLR, 2022.
>
> [6] Zhang, Dinghuai et al. "Generative Flow Networks for Discrete Probabilistic Modeling" International Conference on Machine Learning. PMLR, 2022
>
> [7] Hinton, Geoffrey "Training Products of Experts by Minimizing Contrastive Divergence", Neural Computation 2002

---

### Meta-Review · Area_Chair_zxyp · 2023-12-07

**Metareview:**

The paper introduces a novel approach to training discrete Energy-Based Models (EBMs) using Energy Discrepancy, which simplifies the training process by requiring only the evaluation of the energy function at data points and their perturbed counterparts. This eliminates the need for the Markov chain Monte Carlo. The proposed methods are demonstrated in various tasks, including Ising models training, discrete density estimation, graph generation, and discrete image modeling. While the paper is novel and the results are promising, there are concerns about the weak empirical results. Additionally, certain experiments, such as those involving non-binary cases, need to be  considered. The rebuttal has addressed some concerns, but issues related to scalability and missing performance in the non-binary case remain. The Area Chair recommends rejecting the paper at the current stage and encourages the authors to enhance their paper by incorporating all the valuable suggestions provided by the reviewers, and resubmit it to the next venue.

**Justification For Why Not Higher Score:**

The current experimental results in the paper are not sufficient to fully demonstrate the performance of the method. Thus, the AC rejected it.

**Justification For Why Not Lower Score:**

NA

---

### Decision · Program_Chairs · 2024-01-16

Reject